# Introspective Experience Replay: Look Back When Surprised

**Ramnath Kumar**  *ramnathk@google.com*
*Google Research*

**Dheeraj Nagaraj**  *dheerajnagaraj@google.com*
*Google Research*

**Reviewed on OpenReview:** *https://openreview.net/forum?id=vWTZO1RXZR*

## Abstract

In reinforcement learning (RL), experience replay-based sampling techniques are crucial in promoting convergence by eliminating spurious correlations. However, widely used methods such as uniform experience replay (UER) and prioritized experience replay (PER) have been shown to have sub-optimal convergence and high seed sensitivity, respectively. To address these issues, we propose a novel approach called Introspective Experience Replay (IER) that selectively samples batches of data points prior to surprising events. Our method is inspired from the reverse experience replay (RER) technique, which has been shown to reduce bias in the output of Q-learning-type algorithms with linear function approximation. However, RER is not always practically reliable when using neural function approximation. Through empirical evaluations, we demonstrate that IER with neural function approximation yields reliable and superior performance compared to UER, PER, and hindsight experience replay (HER) across most tasks.

## 1 Introduction

Reinforcement learning (RL) involves learning with dependent data derived from trajectories of Markov processes. In this setting, the iterations of descent algorithms (like SGD) designed for i.i.d. data co-evolve with the trajectory at hand, leading to poor convergence[1]. Experience replay (Lin, 1992) involves storing the received data points in a large buffer and producing a random sample from this buffer whenever the learning algorithm requires it. Therefore, experience replay is usually deployed with popular algorithms like DQN, DDPG, and TD3 to achieve state-of-the-art performance (Mnih et al., 2015; Lillicrap et al., 2015). It has been shown experimentally (Mnih et al., 2015) and theoretically (Nagaraj et al., 2020) that these learning algorithms for Markovian data show sub-par performance without experience replay.

The simplest and most widely used experience replay method is the uniform experience replay (UER), where the data points stored in the buffer are sampled uniformly at random every time a data point is queried (Mnih et al., 2015). However, UER might pick uninformative data points most of the time, which may slow down the convergence. For this reason, optimistic experience replay (OER) and prioritized experience replay (PER) (Schaul et al., 2015) were introduced, where samples with higher TD error (i.e., 'surprise') are sampled more often from the buffer. Optimistic experience replay (originally called "greedy TD-error prioritization" and introduced in 'Schaul et al. (2015)) was shown to have a high bias[2]. This leads to the algorithm predominantly selecting rare data points and ignoring the rest, which are necessary to learn how to

---

[1]The term here is used from the lens of empirical study, where the learning curves do not improve with further training, and the model is said to have converged.

[2]We use the term 'bias' here as used in Schaul et al. (2015), which means biased with respect to the empirical distribution over the replay buffer

reach the high reward states. Prioritized experience replay was proposed to solve this issue (Schaul et al., 2015) by using a sophisticated sampling approach. However, as shown in our experiments outside of the Atari environments, PER still suffers from a similar problem, and its performance can be highly sensitive to seed and hyper-parameter. The design of experience replay continues to be an active field of research. Several other experience replay techniques like Hindsight experience replay (HER) (Andrychowicz et al., 2017), Reverse Experience Replay (RER) (Rotinov, 2019), and Topological Experience Replay (TER) (Hong et al., 2022) have been proposed. An overview of these methods is discussed in Section 2.

Even though these methods are widely deployed in practice, theoretical analyses have been very limited. Recent results on learning dynamical systems (Kowshik et al., 2021b;a) showed rigorously in a theoretical setting that RER is the conceptually-grounded algorithm when learning from Markovian data. Furthermore, this work was extended to the RL setting in Agarwal et al. (2021) to achieve efficient Q learning with linear function approximation. The RER technique achieves good performance since reverse order sampling of the data points prevents the build-up of spurious correlations in the learning algorithm. In this paper, we build on this line of work and introduce **Introspective Experience Replay** (IER). Roughly speaking, IER first picks top $k$ 'pivot' points from a large buffer according to their TD error. It then returns batches of data formed by selecting the consecutive points *temporally* before these pivot points. In essence, the algorithm *looks back when surprised.* The intuition behind our approach is linked to the fact that the agent should always associate outcomes to its past actions, just like in RER. The summary of our approach is shown in Figure 1. This technique is an amalgamation of Reverse Experience Replay (RER) and Optimistic Experience Replay (OER), which only picks the points with the highest TD error.

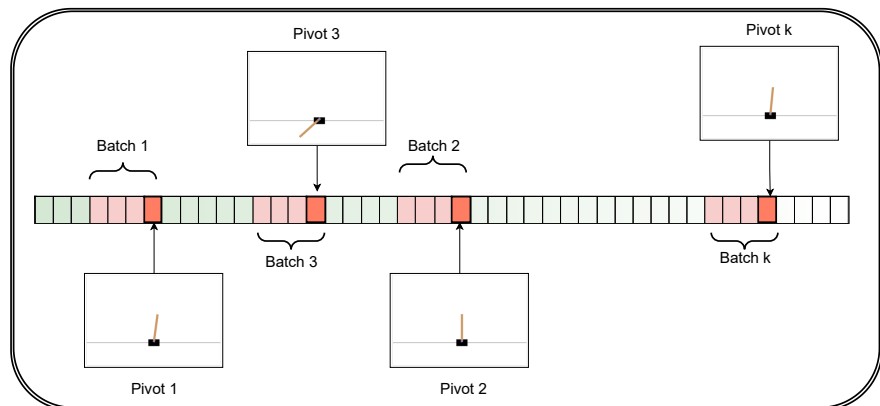

Figure 1: An illustration of our proposed methodology when selecting $k$ batches in the CartPole environment. The red color is used to indicate the batches being sampled from the replay buffer. The green samples are the un-sampled states from the buffer. The arrow explicitly points the pivots and the snapshot of the surprising state encountered.

Our main findings are summarized below:

**Better Performance Against SOTA:** Our proposed methodology (IER) outperforms previous state-of-the-art baselines such as PER, UER and HER on most environments (see Table 1, Section 5).

**Conceptual Understanding:** We consider simple toy examples to understand the differences between UER, RER, OER and IER (Section 3.2, Section 4). This study justifies our algorithm design by showing a) why naive importance sampling, like in OER suffers from poor performance, often failing to learn any non-trivial policy and b) why techniques like UER and RER are slow to learn.

**Forward vs. Reverse:** We show that the temporal direction (forward/reverse) to consider after picking the pivot is non-trivial. We show empirically that IER performs much better than its forward counterpart, IER (F) (see Figure 13). This gives evidence that causality plays a role in the success of IER.

**Whole vs. Component Parts:** Our method (IER) is obtained by combining RER (which picks the samples in the reverse order as received) and OER (which greedily picks the samples with the largest TD

error). However, neither of these components perform well compared to their amalgamation, IER (see Figure 9).

**Minimal hyperparameter tuning:** Our proposed methodology uses minimal hyperparameter tuning. We use the same policy network architecture, learning rate, batch size, and all other parameters across all our runs for a given environment. These hyperparameters are selected based on the setting required to achieve SOTA performance on UER. However, we introduce two hyperparameters: a *Hindsight* flag, and the Uniform Mixing Fraction $p$. When the *Hindsight* parameter is set, then our samples combines with the HER sampler leading to H-IER. The parameter $p$ is the fraction of batches picked as per UER instead of IER in a given buffer (i.e, it allows mixing with UER, giving us U-IER). Note that most of our experiments use $p = 0$.

Table 1: **IER performance in comparison to previous state-of-the-art baselines**. These baselines include samplers such as UER (Mnih et al., 2013), PER (Schaul et al., 2015), and HER (Andrychowicz et al., 2017) across many environments. Results are from 11 different environments that cover a broad category of MDPs. These include a few Atari environments (previously used to support the efficacy of UER, PER), and many other classes of environments, including Classic Control, Box 2D, and Mujoco. More details about these experiments and their setup have been discussed in Section 5.

| Experience Replay Method | UER | PER | HER | IER |
|---|---|---|---|---|
| Best Performance Frequency | 1 | 0 | 0 | 10 |

## 2 Related Works and Comparison

### 2.1 Experience Replay Techniques

Experience replay involves storing consecutive temporally dependent data in a (large) buffer in a FIFO order. Whenever a learning algorithm queries for batched data, the experience replay algorithm returns a sub-sample from this buffer such that this data does not hinder the learning algorithms due to spurious correlations. The most basic form of experience replay is UER (Lin, 1992) which samples the data in the replay buffer uniformly at random. This approach has significantly improved the performance of off-policy RL algorithms like DQN (Mnih et al., 2015). Several other methods of sampling from the buffer have been proposed since; PER (Schaul et al., 2015) samples experiences from a probability distribution which assigns higher probability to experiences with significant TD error and is shown to boost the convergence speed of the algorithm. This outperforms UER in most Atari environments. HER (Andrychowicz et al., 2017) works in the "what if" scenario, where even a bad policy can lead the agent to learn what not to do and nudge the agent towards the correct action. There have also been other approaches such as Liu et al. (2019); Fang et al. (2018; 2019) have adapted HER in order to improve the overall performance with varying intuition. RER processes the data obtained in a buffer in the reverse temporal order. We refer to the following sub-section for a detailed review of this and related techniques. We will also consider 'optimistic experience replay' (OER), the naive version of PER, where at each step, only top $B$ elements in the buffer are returned when batched data is queried. This approach can become highly selective to the point of completely ignoring certain data points, leading to poor performance. This is mitigated by a sophisticated sampling procedure employed in PER. The theoretical analyses for these standard techniques have been explored but are limited (Szlak & Shamir, 2021).

### 2.2 Reverse Sweep Techniques

Reverse sweep or backward value iteration refers to methods that process the data as received in reverse order (see Section 4 for a justification of these methods). This has been studied in the context of planning tabular MDPs where transition probabilites are known (Dai & Hansen, 2007; Grześ & Hoey, 2013) or have access to a simulator or a fitted generative model (Florensa et al., 2017; Moore & Atkeson, 1993; Goyal et al., 2018; Schroecker et al., 2019). However, our work only seeks on-policy access to the MDP.

RER has been shown to work earlier Agarwal et al. (2021) in tabular settings where the state space is significantly small. However, RER has not been shown to scale to settings with neural approximation Rotinov (2019) where most environments have very large state spaces. Therefore, the iterations of RER are 'mixed' with UER. However, the experiments are limited and do not demonstrate that this method outperforms

even UER. A similar procedure named Episodic Backward Update (EBU) is introduced in Lee et al. (2019). However, to ensure that the pure RER works well, the EBU method also seeks to change the target for Q learning instead of just changing the sampling scheme in the replay buffer. The reverse sweep was rediscovered as RER in the context of streaming linear system identification in Kowshik et al. (2021b), where SGD with reverse experience replay was shown to achieve near-optimal performance. In contrast, naive SGD was significantly sub-optimal due to the coupling between the Markovian data and the SGD iterates. The follow-up work Agarwal et al. (2021) analyzed off-policy Q learning with linear function approximation and reverse experience replay to provide near-optimal convergence guarantees using the unique super martingale structure endowed by reverse experience replay. Hong et al. (2022) considers topological experience replay, which executes reverse replay over a directed graph of observed transitions. Mixed with PER enables non-trivial learning in some challenging environments.

## 3 Background and Proposed Methodology

We consider episodic reinforcement learning (Sutton & Barto, 2018), where at each time step $t$ an agent takes actions $a_t$ in an uncertain environment with state $s_t$, and receives a reward $r_t(s_t, a_t)$. The environment then evolves into a new state $s_{t+1}$ whose law depends only on $(s_t, a_t, t)$. Our goal is to (approximately) find the policy $\pi^*$ which maps the environmental state $s$ to an action $a$ such that when the agent takes the action $a_t = \pi^*(s_t)$, the discounted reward $\mathbb{E}\left[\sum_{t=0}^{\infty} \gamma^t r_t\right]$ is maximized. To achieve this, we consider algorithms like DQN (Mnih et al. (2015)), and TD3 (Fujimoto et al. (2018)), which routinely use experience replay buffers. In this paper, we introduce a new experience replay method, IER, and investigate the performance of the aforementioned RL algorithms with this modification. In this work, when we say "return", we mean discounted episodic reward.

---

**Algorithm 1:** Our proposed Introspective Experience Replay (IER) for Reinforcement Learning

---

**Input:** Data collection mechanism $\mathbb{T}$, Data buffer $\mathcal{H}$, Batch size $B$, grad steps per Epoch $G$, number of
episodes $N$, Importance function $I$, learning procedure $\mathbb{A}$, Uniform Mixing fraction $p$

$n \leftarrow 0$
**while** $n < N$ **do**
    $n \leftarrow n + 1$
    $\mathcal{H} \leftarrow \mathbb{T}(\mathcal{H})$                      // Add a new episode to the buffer
    $\mathcal{I} \leftarrow I(\mathcal{H})$           // Compute importance of each data point in the buffer
    $P \leftarrow \mathsf{Top}(\mathcal{I}; G)$                // Obtain index of top $G$ elements of $\mathcal{I}$
    $g \leftarrow 0$
    **while** $g < G$ **do**
        **if** $g < (1 - p)G$ **then**
            $D \leftarrow \mathcal{H}[P[g] - B, P[g]]$       // Load batch of previous $B$ examples from pivot $P[g]$
        **else**
            $D \leftarrow \mathcal{H}[\mathsf{Uniform}(\mathcal{H}, B)]$        // Randomly choose $B$ indices from buffer
        **end**
        $g \leftarrow g + 1$
        $\mathbb{A}(D)$                 // Run the learning algorithm with batch data $D$
    **end**
**end**

---

### 3.1 Methodology

We now describe our main method in a general way where we assume that we have access to a data collection mechanism $\mathbb{T}$ which samples new data points. This then appends the sampled data points to a buffer $\mathcal{H}$ and discards some older data points. The goal is to run an iterative learning algorithm $\mathbb{A}$, which learns from batched data of batch size $B$ in every iteration. We also consider an important metric $I$ associated with the problem. At each step, the data collection mechanism $\mathbb{T}$ collects a new episode and appends it to

the buffer $\mathcal{H}$ and discards some old data points, giving us the new buffer as $\mathcal{H} \leftarrow \mathbb{T}(\mathcal{H})$. We then sort the entries of $\mathcal{H}$ based on the importance metric $I$ and store the indices of the top $G$ data points in an array $P = [P[0], \ldots, P[G-1]]$. Then for every index $(i)$ in $P$, we run the learning algorithm $\mathbb{A}$ with the batch $D = (\mathcal{H}(P[i]), \ldots, \mathcal{H}(P[i] - B + 1))$. In some cases, we can 'mix'[3] this with the standard UER sampling mechanism to reduce bias of the stochastic gradients with respect to the empirical distribution in the buffer, as shown below. Our experiments show that this amalgamation helps convergence in certain cases. We describe this procedure in Algorithm 1.

In the reinforcement learning setting, $\mathbb{T}$ runs an environment episode with the current policy and appends the transitions and corresponding rewards to the buffer $\mathcal{H}$ in the FIFO order, maintaining a total of $1E6$ data points, usually. We choose $\mathbb{A}$ to be an RL algorithm like TD3 or DQN. The importance function $I$ is the magnitude of the TD error with respect to the current Q-value estimate provided by the algorithm $\mathbb{A}$ (i.e., $I = |Q(s,a) - R(s,a) - \gamma \sup_{a'} Q^{\text{target}}(s', a')|$). When the data collection mechanism ($\mathbb{T}$) is the same as in UER, we will call this method **IER**. In optimistic experience replay (OER), we take $\mathbb{T}$ to be the same as in UER. However, we query top $BG$ data points from the buffer $\mathcal{H}$ and return $G$ disjoint batches each of size $B$ from these 'important' points. It is clear that IER is a combination of OER and RER. Notice that we can also consider the data collection mechanism like that of HER, where examples are labeled with different goals, i.e. $\mathbb{T}$ has now been made different, keeping the sampling process exactly same as before. In this case, we will call our algorithm **H-IER**. Our experiment in Enduro and Acrobat depicts an example of this successful coalition. We also consider the RER method, which served as a motivation for our proposed approach. Under this sampling methodology, the batches are drawn from $\mathcal{H}$ in the temporally reverse direction. This approach is explored in the works mentioned in Section 2.2. We discuss this methodology in more detail in Appendix F.

## 3.2 Didactic Toy Example

In this section, we discuss the working of IER on a simple environment such as GridWorld-1D, and compare this with some of our baselines such as UER, OER, RER, and IER (F). In this environment, the agent lives on a discrete 1-dimensional grid of size 40 with a max-timestep of 1000 steps, and at each time step, the agent can either move left or right by one step. The agent starts from the *starting state* (S; [6]), the goal of the agent is to reach *goal state* (G; [40]) getting a reward of $+1$, and there is also a *trap state* (T; [3]), where the agents gets a reward of $-2$. The reward in every other state is 0. For simplicity, we execute an offline exploratory policy where the agent moves left or right with a probability of half and obtain a buffer of size 30000. The rewarding states occur very rarely in the buffer since it is hard to reach for this exploration policy. The episode ends upon meeting either of two conditions: (i) the agent reaches the terminal state, which is the *goal state*, or (ii) the agent has exhausted the max-timestep condition and has not succeeded in reaching any terminal state. An overview of our toy environment is depicted in Figure 2(a). Other hyperparameters crucial to replicating this experiment are described in Appendix B.

In this example, reaching the goal state as quickly as possible is vital to receive a positive reward and avoid the fail state. Therefore, it is essential to understand the paths which reach the goal state. Figure 2(b) depicts the number of times each state occurs in the buffer. Furthermore, the remaining subplots of Figure 2 depict the *Absolute Frequency* of our off-policy algorithm trained in this environment. A state's "absolute frequency" is the number of times the replay technique samples a given state during the algorithm's run. The experiments on this simple didactic toy environment do highlight a few interesting properties:

**Comparison of UER and IER:** Since the goal state appears very rarely in buffer, UER and RER rarely sample the goal state and hence do not manage to learn effectively. While RER naturally propagates the information about the reward back in time to the states that led to the reward, it does not often sample the rewarding state.

**Limitation of OER:** While OER samples a lot from the states close to the goal state, the information about the reward does not propagate to the start state. We refer to the bottleneck in Figure 2(e) where some intermediate states are not sampled.

**Advantage of IER:** IER prioritizes sampling from the goal state and propagates the reward backward so that the entire path leading to the reward is now aware of how to reach the reward. Therefore, a combination

---

[3]Mixing here denotes sampling with a given probability from one sampler A, and filling the remaining samples of a batch with sampler B.

of RER and OER reduces the sampling bias in OER by preventing the bottlenecks seen in Figure 2(e).
**Bottleneck of IER (F):** IER (F) has a more significant bottleneck when compared to RER and chooses to sample the non-rewarding middle states most often. IER (F) picks states in the future of the goal state, which might not contain information about reaching the goal.

This does not allow the algorithm to effectively learn the path which *led* to the goal state.

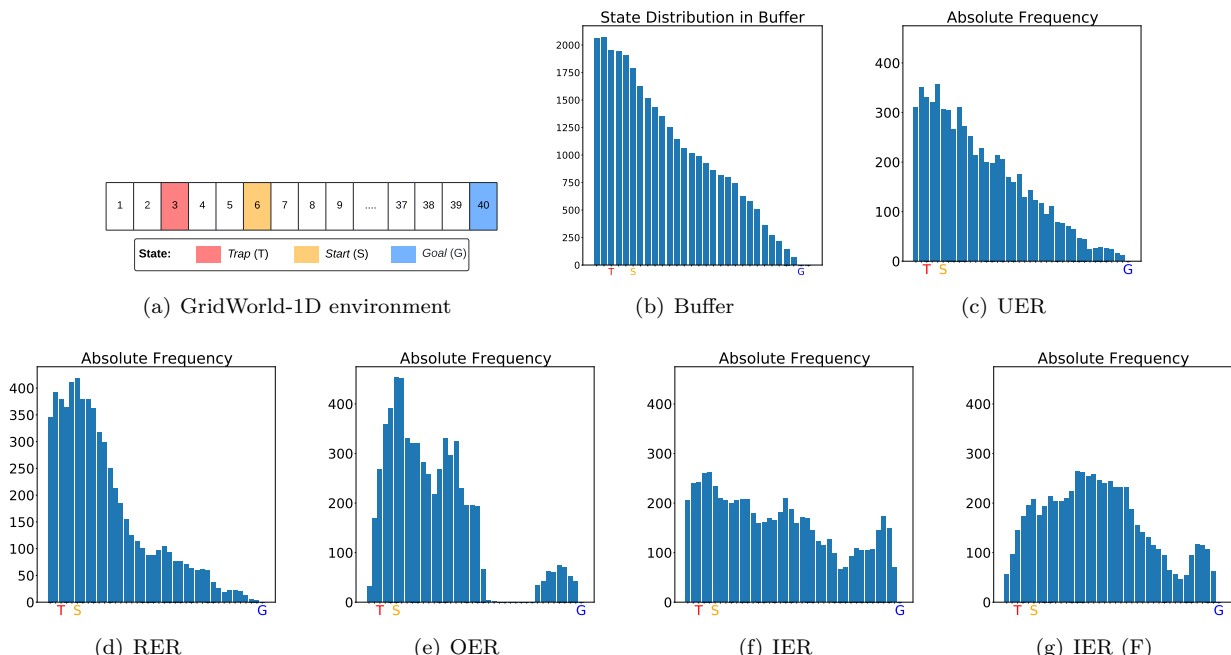

Figure 2: Gridworld-1D environment is depicted in Figure 2(a). Distribution of states in the buffer (Figure 2(b)) and relative frequency of different experience replay samplers on the didactic example of GridWorld-1D environment (Figure 2(c);2(d);2(e);2(f);2(g)).

The toy example above models several salient features in more complicated RL environments.

(i) In the initial stages of learning, the exploratory policy is essentially random, and such a naive exploratory policy does not often lead to non-trivial rewards.

(ii) Large positive and negative reward states (the goal and trap states), and their neighbors provide the pivot state for IER and OER.

We show empirically that this holds in more complicated environments as well. Figure 3 depicts the surprise vs. reward for the Ant environment. Here we see a strong correlation between absolute reward and TD error ("Surprise factor").

## 4 Understanding Reverse Replay

This section highlights some conceptual motivations behind RER and IER. Works such as Agarwal et al. (2021); Kowshik et al. (2021b) have established rigorous theoretical guarantees for RER by utilizing super martingale structures. This structure is not present in forward replay techniques (i.e., the opposite of reverse replay) as shown in Kowshik et al. (2021a). We refer to Appendix D, where we show via an ablation study that going forward in time instead of reverse does not work very well. We give the following explanations for the success of IER, but note that further theoretical investigation is needed.
**Propogation of Sparse Rewards:** In many RL problems, non-trivial rewards are sparse and only received at the goal states. Therefore, processing the data backward in time from such goal states helps the algorithm learn about the states that led to this non-trivial reward. Our study (see Figure 3 and Appendix E for

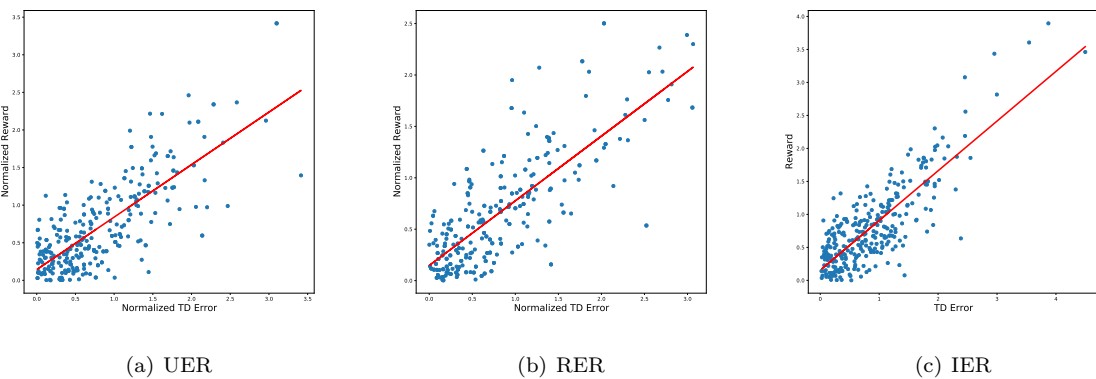

| (a) UER | (b) RER | (c) IER |

Figure 3: Relationship between absolute values of TD Error (Surprise factor) and Reward for the Ant environment.

further details) shows that in many environments IER picks pivots which are the states with large (positive or negative) rewards, enabling effective learning.

**More spread than OER:** OER, which greedily chooses the examples with the largest TD error to learn from, performs very poorly since it is overtly selective to the point of ignoring most of the states. To illustrate this phenomenon, we refer to the didactic example in Section 3.2. One possible way of viewing IER is that RER is used to reduce this affinity to picking a minimal subset of states in OER. PER is designed to achieve a similar outcome with a sophisticated and computationally expensive sampling scheme over the buffer.

**Causality:** MDPs have a natural causal structure: actions and events in the past influence the events in the future. Therefore, whenever we see a surprising or unexpected event, we can understand why or how it happened by looking into the past.

We further illustrate the same by referring to the straightforward didactic example (Section 3.2), where we can see the effects of each of the experience replay methods. We also demonstrate superior performance on more complicated environments (Section 5), showcasing the robustness of our approach with minimal hyperparameter tuning.

### 4.1 Results on Chain Environment

In this section, we discuss the working of IER on two simple Markov chain environments as proposed by Zhang & Yao (2019), where the environments were used for illustrating the inefficiency of mean-based decision-making. Similar to the didactic toy example (Section 3.2), we compare some of our baselines, such as UER, OER, and RER to further show the efficacy and robustness of our proposed approach. Our setup of the environment is identical to the setup proposed in Figure 1 of Zhang & Yao (2019) paper. Briefly, we experiment on two Markov Chain problems as described below:

Chain 1 is a system with N non-terminal states and two available actions, UP and RIGHT. At the beginning of each episode, the agent starts at state 1. The UP action results in an immediate end to the episode with no reward, while the RIGHT action leads to the next state with a reward sampled from a normal distribution with a mean of 0 and standard deviation of 1. When the agent reaches the terminal state G, the episode ends with a reward of +10, and there is no discounting. The optimal policy for this system is always moving right, but to learn this policy, the agent must first reach the G state. However, this is challenging for Q-learning because of two factors: the epsilon-greedy mechanism may randomly select UP, ending the episode, and before reaching G, the expected return of either RIGHT or UP at any state is 0, making it impossible for the agent to differentiate between the two actions based on the mean criterion.

Chain 2 has the same state and action space as Chain 1, but the reward for RIGHT is now -0.1, except for reaching the G state, which gives a reward of +10. The reward for UP is sampled from a normal distribution with a mean of 0 and a standard deviation of 0.2. There is no discounting. When the standard deviation

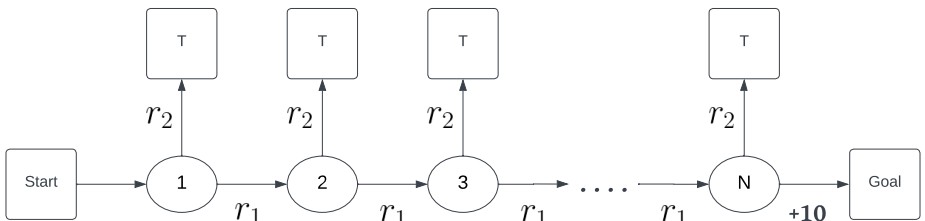

Figure 4: An illustration of the chain environment used in our below experiments. In Markov chain 1, we set $r_1 = \mathcal{N}(0, 1)$, and $r_2 = 0$. In Markov chain 2, we set $r_1 = -0.1$, and $r_2 = \mathcal{N}(0, 0.2)$.

is small, the optimal policy remains moving right. However, before reaching the G state, the expected return of RIGHT for any non-terminal state is less than 0, which means a Q-learning agent would prefer UP. This preference is detrimental to Chain 2 as it leads to an immediate episode ending, preventing further exploration.

Furthermore, to make the problem harder (in comparison to the works of Zhang & Yao (2019)), we have increased the number of non-terminal states (N) to 10. We train the agent with a warmup of 1000 steps to populate the replay buffer similar to the setup in Section 3.2. Additional hyperparameters used for our agent are described in Table 5. We report the number of epochs required to reach the optimal policy averaged over 50 different seeds, with a maximum training epoch limit of 100. The lower the number of epochs taken for convergence, the better. As presented in the Table 2, our approach of utilizing IER outperforms other methods by effectively capturing the idea of trap states and demonstrates superior learning capabilities.

Table 2: Number of epochs required to reach the optimal policy averaged over 50 different seeds. From our experiments, we note that IER outperforms previous SOTA baselines in both environments. The lower the number of epochs taken for convergence, the better.

| Environment | UER | HER | RER | OER | IER | IER-F |
|---|---|---|---|---|---|---|
| Markov Chain 1 | $49.52_{\pm 24.65}$ | $62.22_{\pm 29.47}$ | $45.72_{\pm 28.04}$ | $7.36_{\pm 1.67}$ | $\mathbf{4.22}_{\pm 1.51}$ | $8.64_{\pm 1.86}$ |
| Markov Chain 2 | $96.98_{\pm 5.16}$ | $98.76_{\pm 1.24}$ | $97.06_{\pm 5.41}$ | $8.04_{\pm 1.23}$ | $\mathbf{6.72}_{\pm 0.57}$ | $15.82_{\pm 1.28}$ |

## 5  Experimental Results

In this section, we briefly discuss our experimental setup as well as the results of our experiments.

**Environments:** We evaluate our approach on a diverse class of environments, such as (i) Environments with low-dimensional state space (including classic control and Box-2D environments), (ii) Multiple joint dynamic simulations environments (including Mujoco environments), and (iii) Human-challenging environments (such as Atari environments). Note that previous seminal papers in the field of experience replay, such as Mnih et al. (2013), Schaul et al. (2015), and Andrychowicz et al. (2017), showed the efficacy of their approach on a subset of these classes. For instance, UER and PER was shown to work well on Atari games. In this work, we perform an extensive study to show the robustness and effectiveness of our model not just in Atari environments but also in Mujoco, Box2D, and Classic Control environments. Due to computational limitations and the non-reproducibility of some baselines with UER, we could not extend our experiments to some environments. We refer to Appendix A for a brief description of the environments used.

**Hyperparameters:** Refer to Appendix B for the exact hyperparameters used. Across all our experiments on various environments, we use a standard setting for all the different experience replay buffers. This classic setting is set so we can reproduce state-of-the-art performance using UER on the respective environment. For most of our experiments, we set the uniform mixing fraction ($p$) from Algorithm 1 to be 0. We use a

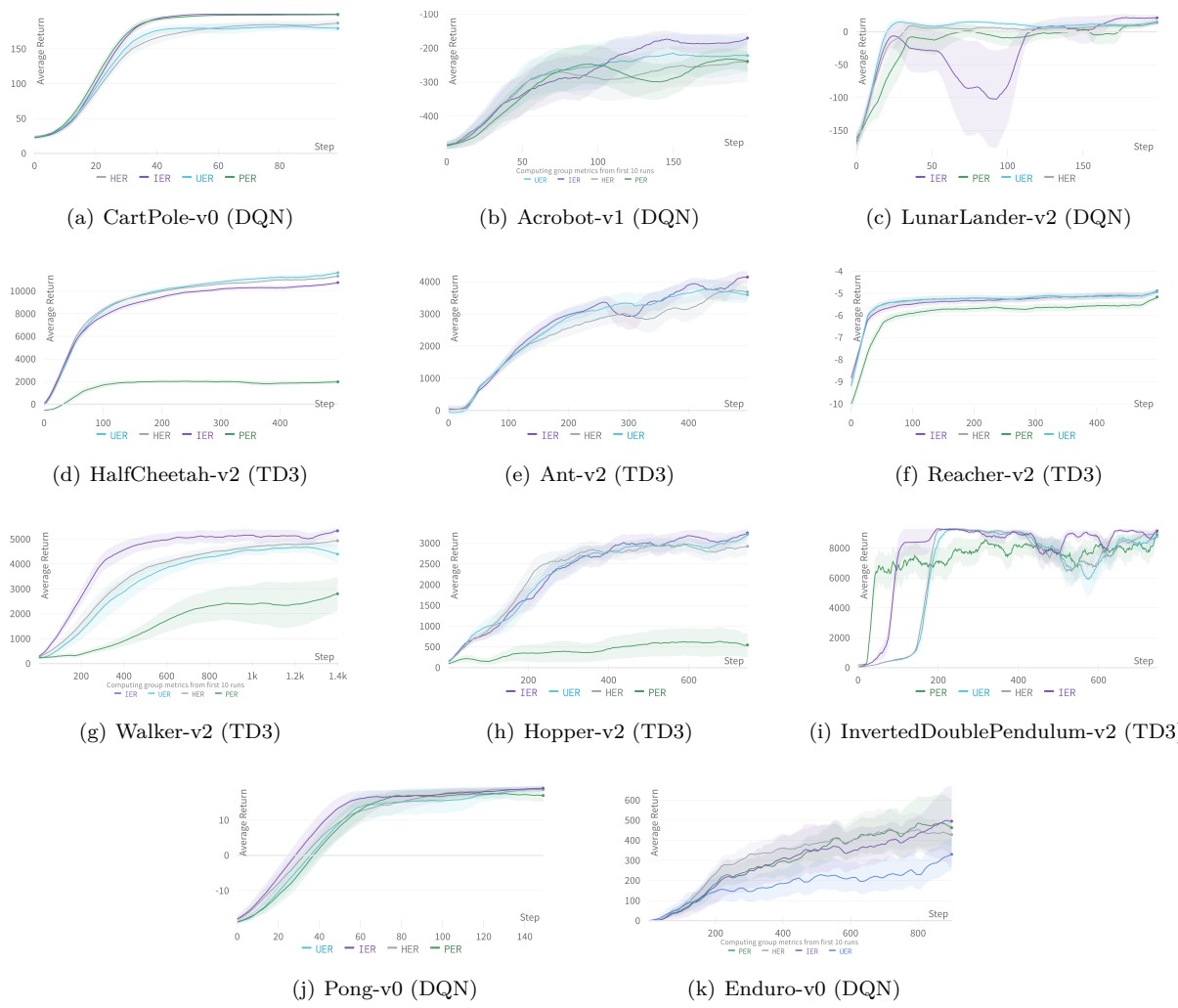

Figure 5: Learning curves of DQN/TD3 agents on various environments using the Top-k metric. Here, the shaded area denotes the standard error.

non-zero $p$ value only for a few environments to avoid becoming overtly selective while training, as described in Appendix B. For PER, we tune the $\alpha$ and $\beta$ hyperparameters used in the Schaul et al. (2015) paper across all environments other than Atari. The default values of $\alpha = 0.4$ and $\beta = 0.6$ are robust on Atari environments as shown by extensive hyperparameter search by Schaul et al. (2015). We detail the results from our grid search in Appendix C.2.

**Metric:** The metric for comparison between different algorithms is very important in RL since the learning curves are extremely noisy, seed sensitive and contain many outliers. Thus the empirical mean estimation method (where the empirical average of the return is taken across the seeds) might not be able to compare the performance of the algorithms with a reasonable number of samples. To mitigate this drawback, we report our results with respect to two metrics (defined in the paragraphs below): The robust mean (the statistically principled approach) and the top-K seeds moving average return (heavily used in prior works). We demonstrate the robustness of the top-K metric and the robust mean metric from our study in Appendix C.1. Here the task is to infer the best algorithm based on a toy model of highly seed-sensitive learning curves. We show that both these methods allow us to efficiently infer the best algorithm. However, the usual empirical mean chooses the wrong answer almost 50% of the time. The top-K metric has been criticized in the literature since this can empirically pick relatively sub-optimal algorithms which perform very well on rare seeds (Henderson et al., 2018). Thus, while we primarily consider the robust mean metric to compare the

algorithms, we also report the top-K metric in order to facilitate a comparison with prior works. We now describe the metrics used in detail below.

**Robust Mean Metric:** Practical data is known to suffer from heavy tailed phenomenon and outliers and thus estimating their means with the empirical mean estimator can be very sub-optimal. Thus, robust estimation has been widely studied in statistics, going back to the works of Huber (1964) and Tukey (1975). We refer to Lugosi & Mendelson (2019) for a detailed literature review on this topic. Thus we consider the median of means estimator (see refs in Lugosi & Mendelson (2019)) as the robust estimator for the mean. We consider $n = 25$ seeds for each environment and each algorithm ($n = 10$ for the Atari environments due to resource constraints) and plot the moving average with a window size of 20 for CartPole and 50 for all others. We divide the moving average reward into 5 buckets of equal size and compute the mean inside each bucket. We then output the median across of these means. We construct the standard error of this estimator using the popular bootstrap method Efron & Tibshirani (1994).

**The Top-K Seeds Metric:** It is common to use Top-1 and Top-K trials to be selected from among many trials in the reinforcement learning literature (see Schaul et al. (2015);Sarmad et al. (2019);Wu et al. (2017);Mnih et al. (2016)). In our work, we use the *Top-K seeds moving average return* as the evaluation metric across all our runs. Top-K seeds here mean we take the average of $k = 10$ seeds that gave the best performance out of a total of $n = 25$ trials. This measures the average performance of the top $k/n$ quantile of the trajectories. This is robust despite the highly stochastic learning curves encountered in RL, where there is a non-zero probability that non-trivial policies are not learned after many steps. Moving average with a given window size is taken for learning curves (with a window size of 20 for CartPole and 50 for all others) to reduce the variation in return which is inherently present in each epoch. We argue that taking a moving average is essential since, usually, pure noise can be leveraged to pick a time instant where a given method performs best (Henderson et al., 2018). Considering the Top-K seed averaging of the last step performance of the moving average of the learning curves gives our metric - the *Top-K seeds moving average return.*

**Comparison with SOTA:** Figure 5 and Figure 6 depicts our results in various environments upon using different SOTA replay sampler mechanisms (UER, PER and HER) on the *Top-k* and *Robust Mean* metric. Our proposed sampler outperforms all other baselines in most tasks and compares favorably in others. Our experiments on various environments across various classes, such as classic control, and Atari, amongst many others, show that our proposed methodology consistently outperforms all other baselines in most environments. Furthermore, our proposed methodology is robust across various environments. (See Table 3)

**Forward vs. Reverse:** The intuitive limitation to the "looking forward" approach is that in many RL problems, the objective for the agent is to reach a final goal state, where the non-trivial reward is obtained. Since non-trivial rewards are only offered in this goal state, it is informative to look back from here to learn about the states that *lead* to this. When the goals are sparse, the TD error is more likely to be large upon reaching the goal state. Our algorithm selects these as pivots, and IER (F) picks states in the future of the goal state, which might not contain information about reaching the goal. Our studies on many environments (see Figure 3 and Appendix E) show that the pivot points selected based on importance indeed have large (positive or negative) rewards.

**Whole vs. Component Parts:** Our approach is an amalgamation of OER and RER. Here we compare these individual parts with IER. Figure 9 describes the *Top-K seeds Moving Average Return* across various environments in this domain. As demonstrated, IER outperforms its component parts OER and RER . Furthermore, we also motivate each of our design choices such as the pivot point selection (see Appendix C.5 where we compare our proposed approach with other variants of IER where the pivot points are randomly selected), and temporal structure (see Appendix C.6 where we compare our proposed approach with other variants of IER where the points are randomly sampled from the buffer instead of temporally looking backward after selecting a pivot point), and also buffer batch size sensitivity (see Appendix C.4).

## 6 Discussion

We summarize our results and discuss possible future steps.

**Speedup:** IER shows a significant speedup in terms of time complexity over PER as depicted in Table 4. On average IER achieves a speedup improvement of 26.20% over PER across a large umbrella of environment classes. As the network becomes more extensive or complicated, our approach does have a higher overhead (especially computing TD error). Future work can investigate how to further reduce the computational

Table 3: Average reward (via robust mean estimation) across various environments along with the standard error (computed via bootstrap). From our experiments, we note that IER outperforms previous SOTA baselines in most environments. Appendix B depicts the hyperparameters used for the experiment.

| Dataset | *UER* | *PER* | *HER* | *IER* |
|---|---|---|---|---|
| CartPole | $170.98 _{\pm 8.05}$ | $177.55 _{\pm 10.36}$ | $170.16 _{\pm 8.06}$ | $\mathbf{199.19} _{\pm 0.86}$ |
| Acrobot | $-347.31 _{\pm 74.45}$ | $-339.08 _{\pm 48.85}$ | $-328.36 _{\pm 70.39}$ | $\mathbf{-256.62} _{\pm 55.1}$ |
| LunarLander | $1.62 _{\pm 4.03}$ | $-0.34 _{\pm 6.06}$ | $5.45 _{\pm 3.2}$ | $\mathbf{11.16} _{\pm 9.74}$ |
| HalfCheetah | $\mathbf{11047.69} _{\pm 79.56}$ | $404.09 _{\pm 288.65}$ | $10984.45 _{\pm 123.32}$ | $10179.05 _{\pm 114.84}$ |
| Ant | $3576.19 _{\pm 271.51}$ | $-2700.74 _{\pm 0.36}$ | $3688.03 _{\pm 268.35}$ | $\mathbf{3697.42} _{\pm 293.51}$ |
| Reacher | $-5.11 _{\pm 0.02}$ | $-5.58 _{\pm 0.03}$ | $-5.12 _{\pm 0.02}$ | $\mathbf{-5.09} _{\pm 0.02}$ |
| Walker | $4400.33 _{\pm 168.33}$ | $1120.02 _{\pm 435.17}$ | $4452.59 _{\pm 131.3}$ | $\mathbf{4705.08} _{\pm 174.11}$ |
| Hopper | $2886.62 _{\pm 166.07}$ | $11.14 _{\pm 109.83}$ | $2820.83 _{\pm 311.27}$ | $\mathbf{3221.48} _{\pm 44.81}$ |
| Inverted Double Pendulum | $6876.1 _{\pm 947.31}$ | $7822.22 _{\pm 262.64}$ | $7118.51 _{\pm 1026.44}$ | $\mathbf{7916.32} _{\pm 777.53}$ |
| Pong | $17.47 _{\pm 0.85}$ | $17.67 _{\pm 0.25}$ | $18.17 _{\pm 0.47}$ | $\mathbf{18.44} _{\pm 0.37}$ |
| Enduro | $345.36 _{\pm 72.86}$ | $381.45 _{\pm 71.26}$ | $421.52 _{\pm 30.83}$ | $\mathbf{465.94} _{\pm 25.39}$ |

complexity of our method by computing the TD error fewer times at the cost of operating with an older TD error. We also notice a speedup of convergence toward a local-optimal policy of our proposed approach, as shown in a few environments. Furthermore, the lack of speedup in some of the other experiments (even if they offer an overall performance improvement) could be since the "surprised" pivot cannot be successfully utilized to teach the agent rapidly in the initial stages.

**Issues with stability and consistency.** Picking pivot points by looking at the TD error might cause us to sample rare events much more often and hence cause instability compared to UER as seen in some environments like HalfCheetah, LunarLander, and Ant, where there is a sudden drop in performance for some episodes (see Figure 5, and Figure 6). We observe that our strategy IER corrects itself quickly, unlike RER, which cannot do this (see Figure 9(h)). Increasing the number of pivot points per episode (the parameter $G$) and the uniform mixing probability $p$ usually mitigates this. In this work, we do not focus on studying the exact effects of $p$ and $G$ since our objective was to obtain methods that require minimal hyper-parameter tuning. However, future work can systematically investigate the significance of these parameters in various environments.

Table 4: Average Speedup in terms of time complexity over PER across various environment classes.

| Environment | *Average Speedup* |
|---|---|
| Classic Control | 32.66% ↑ |
| Box-2D | 54.32% ↑ |
| Mujoco | 18.09% ↓ |
| Atari | 6.53% ↑ |

**Why does IER outperform the traditional RER?** The instability[4] and unreliability of pure RER with neural approximation has been noted in various works (Rotinov, 2019; Lee et al., 2019), where RER is stabilized by mixing it with UER . Hong et al. (2022) stabilizes reverse sweep by mixing it with PER. This is an interesting phenomenon since RER is near-optimal in the tabular and linear approximation settings (Agarwal et al., 2021). Two explanations of this are i) The work on RER (Agarwal et al., 2021) considers the linear function approximation setting, where SGD type algorithms with i.i.d. have a bias-variance decomposition Jain et al. (2018): the distance from the initial point contracts exponentially in time (the bias term) and a remainder zero mean term (the variance term). With data from a Markov chain, the variance term can have non-zero mean in the Q learning setting. This problem is mitigated by the RER technique which makes sure the variance term is of zero mean, leading to fast convergence. In the case of non-convex loss functions, such as the square loss over a neural network, such a bias variance decomposition cannot be achieved. Thus, we believe RER does not work well with non-convex loss functions and ii) The proof given in Agarwal et al. (2021) relies extensively on 'coverage' of the entire state-action space - that is, the entire state-action space is visited enough number of times - which might not hold, as shown in Section 3.2.

---

[4]i.e., sudden drop in performance without recovery like catastrophic forgetting

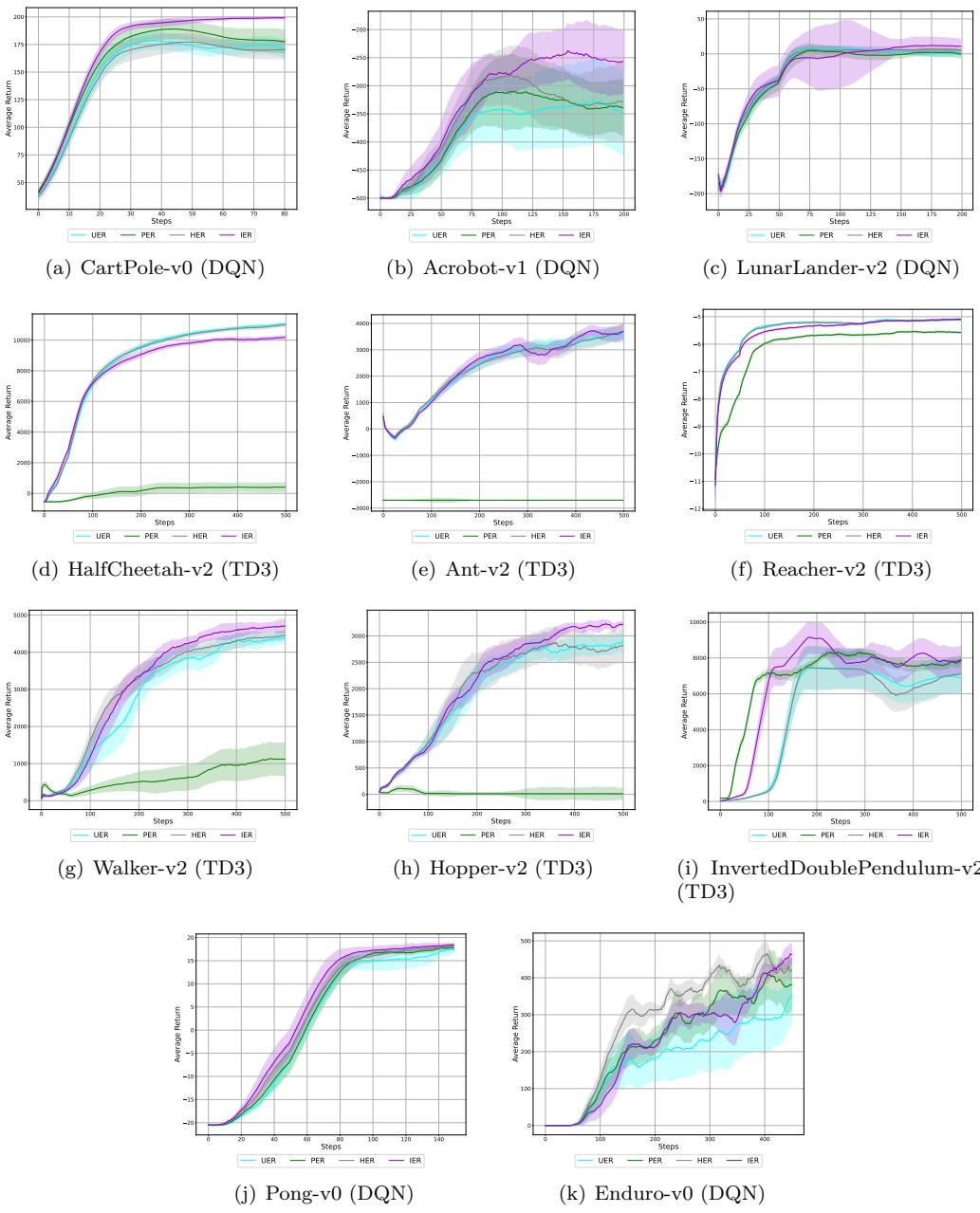

Figure 6: Learning curves of DQN/TD3 agents on various environments using the Robust Mean metric. Here, the shaded area denotes the standard error computed via bootstrap method.

## 7   Conclusion

We propose a novel experience replay sampler, Introspective Experience Replay (IER), which has a significant promise as a solution for improving the convergence of RL algorithms. We have demonstrated through extensive experiments that IER outperforms state-of-the-art techniques such as uniform experience replay, prioritized experience replay, and hindsight experience replay on a wide range of tasks. One of the key strengths of our proposed approach is its ability to selectively sample batches of data points prior to surprising events, which allows fast convergence without incurring the issues in methods such as PER and RER. While IER is constructed based on the theoretically sound RER, it is important to note that IER does not necessarily inherit identical theoretical guarantees.

## Reproducibility Statement

In this paper, we work with thirteen datasets, all of which are open-sourced in gym (https://github.com/openai/gym). More information about the environments is available in Appendix A. We predominantly use DQN, DDPG and TD3 algorithms in our research, both of which have been adapted from their open-source code. We also experimented with seven different replay buffer methodologies, all of which have been adapted from their source code[5]. More details about the models and hyperparameters are described in Appendix B. All runs have been run using the A100-SXM4-40GB, TITAN RTX, and V100 GPUs. Our source code is made available for additional reference [6].

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

# A   Environments

For all OpenAI environments, data is summarized from https://github.com/openai/gym, and more information is provided in the wiki https://github.com/openai/gym/wiki. Below we briefly describe some of the tasks we experimented on in this paper.

## A.1   CartPole-v0

CartPole, as introduced in Barto et al. (1983), is a task of balancing a pole on top of the cart. The cart has access to position and velocity as its state vector. Furthermore, it can go either left or right for each action. The task is over when the agent achieves 200 timesteps without a positive reward (balancing the pole) which is the goal state or has failed, either when (i) the cart goes out of boundaries ($\pm$ 2.4 units off the center), or (ii) the pole falls over (less than $\pm$ 12 deg). The agent is given a continuous 4-dimensional space describing the environment and can respond by returning one of two values, pushing the cart either right or left.

## A.2   Acrobot-v1

Acrobot, as introduced in Sutton (1995), is a task where the agent is given rewards for swinging a double-jointed pendulum up from a stationary position. The agent can actuate the second joint by one of three actions: left, right, or no torque. The agent is given a six-dimensional vector comprising the environment's angles and velocities. The episode terminates when the end of the second pole is over the base. Each timestep that the agent does not reach this state gives a -1 reward, and the episode length is 500 timesteps.

## A.3   LunarLander-v2

The LunarLander environment introduced in Brockman et al. (2016) is a classic rocket trajectory optimization problem. The environment has four discrete actions - do nothing, fire the left orientation engine, fire the right orientation engine, and fire the main engine. This scenario is per Pontryagin's maximum principle, as it is optimal to fire the engine at full throttle or turn it off. The landing coordinates (goal) is always at $(0, 0)$. The coordinates are the first two numbers in the state vector. There are a total of 8 features in the state vector. The episode terminates if (i) the lander crashes, (ii) the lander gets outside the window, or (iii) the lander does not move nor collide with any other body.

## A.4   HalfCheetah-v2

HalfCheetah is an environment based on the work by Wawrzyński (2009) adapted by Todorov et al. (2012). The HalfCheetah is a 2-dimensional robot with nine links and eight joints connecting them (including two paws). The goal is to apply torque on the joints to make the cheetah run forward (right) as fast as possible, with a positive reward allocated based on the distance moved forward and a negative reward is given for moving backward. The torso and head of the cheetah are fixed, and the torque can only be applied to the other six joints over the front and back thighs (connecting to the torso), shins (connecting to the thighs), and feet (connecting to the shins). The reward obtained by the agent is calculated as follows:

$$r_t = \dot{x_t} - 0.1 * \|a_t\|_2^2$$

## A.5   Ant-v2

Ant is an environment based on the work by Schulman et al. (2015) and adapted by Todorov et al. (2012). The ant is a 3D robot with one torso, a free rotational body, and four legs. The task is to coordinate the four legs to move in the forward direction by applying torques on the eight hinges connecting the two links of each leg and the torso. Observations consist of positional values of different body parts of the ant, followed by the velocities of those individual parts (their derivatives), with all the positions ordered before all the velocities. The reward obtained by the agent is calculated as follows:

$$r_t = \dot{x_t} - 0.5 * \|a_t\|_2^2 - 0.0005 * \left\|s_t^{\text{contact}}\right\|_2^2 + 1$$

### A.6 Reacher-v2

The Reacher environment, as introduced in Todorov et al. (2012), is a two-jointed robot arm. The goal is to move the robot's end effector (called *fingertip*) close to a target that is spawned at a random positions. The action space is a two-dimensional vector representing the torque to be applied at the two joints. The state space consists of angular positions (in terms of cosine and sine of the angle formed by the two moving arms), coordinates, and velocity states for different body parts followed by the distance from target for the whole object.

### A.7 Hopper-v2

The Hopper environment, as introduced in Todorov et al. (2012), sets out to increase the number of independent state and control variables compared to classic control environments. The hopper is a two-dimensional figure with one leg that consists of four main body parts - the torso at the top, the thigh in the middle, the leg at the bottom, and a single foot on which the entire body rests. The goal of the environment is to make hops that move in the forward (right) direction by applying torques on the three hinges connecting the body parts. The action space is a three-dimensional element vector. The state space consists of positional values for different body parts followed by the velocity states of individual parts.

### A.8 Walker-v2

The Walker environment, as builds on top of the Hopper environment introduced in Todorov et al. (2012), by adding another set of legs making it possible for the robot to walker forward instead of hop. The hopper is a two-dimensional figure with two legs that consists of four main body part s - the torso at the top, two thighs in the middle, two legs at the bottom, and two feet on which the entire body rests. The goal of the environment is to coordinate both feel and move in the forward (right) direction by applying torques on the six hinges connecting the body parts. The action space is a six-dimensional element vector. The state space consists of positional values for different body parts followed by the velocity states of individual parts.

### A.9 Inverted Double-Pendulum-v2

Inverted Double-Pendulum as introduced in Todorov et al. (2012) is built upon the CartPole environment as introduced in Barto et al. (1983), with the infusion of Mujoco. This environment involves a cart that can be moved linearly, with a pole fixed and a second pole on the other end of the first one (leaving the second pole as the only one with one free end). The cart can be pushed either left or right. The goal is to balance the second pole on top of the first pole, which is on top of the cart, by applying continuous forces on the cart. The agent takes a one-dimensional continuous action space in the range [-1,1], denoting the force applied to the cart and the sign depicting the direction of the force. The state space consists of positional values of different body parts of the pendulum system, followed by the velocities of those individual parts (their derivatives) with all the positions ordered before all the velocities. The goal is to balance the double-inverted pendulum on the cart while maximizing its height off the ground and having minimum disturbance in its velocity.

### A.10 Pong-v0

Pong, also introduced in Mnih et al. (2013), is comparatively more accessible than other Atari games such as Enduro. Pong is a two-dimensional sports game that simulates table tennis. The player controls an in-game paddle by moving vertically across the left and right sides of the screen. Players use this paddle to hit the ball back and forth. The goal is for each player to reach eleven points before the opponent, where the point is earned for each time the agent returns the ball and the opponent misses.

### A.11 Enduro-v0

Enduro, introduced in Mnih et al. (2013), is a hard environment involving maneuvering a race car in the National Enduro, a long-distance endurance race. The goal of the race is to pass a certain number of cars each day. The agent must pass 200 cars on the first day and 300 cars on all subsequent days. Furthermore, as time passes, the visibility changes as well. At night in the game, the player can only see the oncoming cars' taillights. As the days' progress, cars will become more challenging to avoid. Weather and time of day are factors in how to play. During the day, the player may drive through an icy patch on the road, which would limit control of the vehicle, or a patch of fog may reduce visibility.

## B Model and Hyperparameters

The hyperperparameters used for our agents in the Tabular MDP setting with Markov chain environments is shown in Table 5. In this paper, we work with two classes of algorithms: DQN and TD3. The hyperparameters used for training our DQN algorithms in various environments are described in Table 6. Furthermore, the hyperparameters used for training TD3 are described in Table 7.

Table 5: Hyperparameters used for training Tabular MDP on both Markov chain environments.

| Description | Markov chains | argument_name |
|---|---|---|
| Discount | 1 | discount |
| Batch size | 1 | batch_size |
| Number of epochs | 100 | n_epochs |
| Replay Buffer size | $3e^4$ | buffer_size |
| Buffer batch size | 64 | batch_size |
| Exploration factor | 0.3 | max_epsilon |
| Learning rate | 0.1 | lr |

Table 6: Hyperparameters used for training DQN on various environments.

| Description | CartPole | Acrobot | LunarLander | Pong | Enduro | argument_name |
|---|---|---|---|---|---|---|
| *General Settings* | | | | | | |
| Discount | 0.9 | 0.9 | 0.9 | 0.99 | 0.99 | discount |
| Batch size | 512 | 512 | 512 | 32 | 32 | batch_size |
| Number of epochs | 100 | 100 | 200 | 150 | 800 | n_epochs |
| Steps per epochs | 10 | 10 | 10 | 20 | 20 | steps_per_epoch |
| Number of train steps | 500 | 500 | 500 | 125 | 125 | num_train_steps |
| Target update frequency | 30 | 30 | 10 | 2 | 2 | target_update_frequency |
| Replay Buffer size | $1e^6$ | $1e^6$ | $1e^6$ | $1e^4$ | $1e^4$ | buffer_size |
| *Algorithm Settings* | | | | | | |
| CNN Policy Channels | - | - | - | $(32, 64, 64)$ | $(32, 64, 64)$ | cnn_channel |
| CNN Policy Kernels | - | - | - | $(8, 4, 3)$ | $(8, 4, 3)$ | cnn_kernel |
| CNN Policy Strides | - | - | - | $(4, 2, 1)$ | $(4, 2, 1)$ | cnn_stride |
| Policy hidden sizes (MLP) | $(8, 5)$ | $(8, 5)$ | $(8, 5)$ | $(512, )$ | $(512, )$ | pol_hidden_sizes |
| Buffer batch size | 64 | 128 | 128 | 32 | 32 | batch_size |
| *Exploration Settings* | | | | | | |
| Max epsilon | 1.0 | 1.0 | 1.0 | 1.0 | 1.0 | max_epsilon |
| Min epsilon | 0.01 | 0.1 | 0.1 | 0.01 | 0.01 | min_epsilon |
| Decay ratio | 0.4 | 0.4 | 0.12 | 0.1 | 0.1 | decay_ratio |
| *Optimizer Settings* | | | | | | |
| Learning rate | $5e^{-5}$ | $5e^{-5}$ | $5e^{-5}$ | $1e^{-4}$ | $1e^{-4}$ | lr |
| *PER Specific Settings* | | | | | | |
| Prioritization Exponent | 0.4 | 0.6 | 0.8 | 0.4 | 0.4 | $\alpha$ |
| Bias Annealing Parameter | 0.6 | 0.6 | 0.8 | 0.6 | 0.6 | $\beta$ |
| *IER Specific Settings* | | | | | | |
| Use Hindsight for storing states | – | ✓ | – | – | ✓ | use_hindsight |
| Mixing Factor (p) | 0 | 0 | 0 | 0 | 0 | p |

Additionally, we use Tabular MDPs for learning a policy in our toy example. Since the environment is fairly simpler, and has very few states, function approximation is unnecessary. For all the agents trained on GridWorld, we use a common setting as described in Table 8.

Table 7: Hyperparameters used for training TD3 on various environments.

| Description | Ant | Reacher | Walker | Double-Pendulum | HalfCheetah | Hopper | argument_name |
|---|---|---|---|---|---|---|---|
| *General Settings* | | | | | | | |
| Discount | 0.99 | 0.99 | 0.99 | 0.99 | 0.99 | 0.99 | discount |
| Batch size | 250 | 250 | 250 | 100 | 100 | 100 | batch_size |
| Number of epochs | 500 | 500 | 500 | 750 | 500 | 500 | n_epochs |
| Steps per epochs | 40 | 40 | 40 | 40 | 20 | 40 | steps_per_epoch |
| Number of train steps | 50 | 50 | 50 | 1 | 50 | 100 | num_train_steps |
| Replay Buffer size | $1e^6$ | $1e^6$ | $1e^6$ | $1e^6$ | $1e^6$ | $1e^6$ | buffer_size |
| *Algorithm Settings* | | | | | | | |
| Policy hidden sizes (MLP) | $(256, 256)$ | $(256, 256)$ | $(256, 256)$ | $(256, 256)$ | $(256, 256)$ | $(256, 256)$ | pol_hidden_sizes |
| Policy noise clip | 0.5 | 0.5 | 0.5 | 0.5 | 0.5 | 0.5 | pol_noise_clip |
| Policy noise | 0.2 | 0.2 | 0.2 | 0.2 | 0.2 | 0.2 | pol_noise |
| Target update tau | 0.005 | 0.005 | 0.005 | 0.005 | 0.005 | 0.005 | tau |
| Buffer batch size | 100 | 100 | 100 | 100 | 100 | 100 | batch_size |
| *Gaussian noise Exploration Settings* | | | | | | | |
| Max sigma | 0.1 | 0.1 | 0.1 | 0.1 | 0.1 | 0.1 | max_sigma |
| Min sigma | 0.1 | 0.1 | 0.1 | 0.1 | 0.1 | 0.1 | min_sigma |
| *Optimizer Settings* | | | | | | | |
| Policy Learning rate | $1e^{-3}$ | $1e^{-3}$ | $1e^{-4}$ | $3e^{-4}$ | $1e^{-3}$ | $3e^{-4}$ | pol_lr |
| QF Learning rate | $1e^{-3}$ | $1e^{-3}$ | $1e^{-3}$ | $1e^{-3}$ | $1e^{-3}$ | $1e^{-3}$ | qf_lr |
| *PER Specific Settings* | | | | | | | |
| Prioritization Exponent | 0.4 | 0.4 | 0.4 | 0.8 | 0.4 | 0.4 | $\alpha$ |
| Bias Annealing Parameter | 0.6 | 0.6 | 0.6 | 0.6 | 0.6 | 0.2 | $\beta$ |
| *IER Specific Settings* | | | | | | | |
| Use Hindsight for storing states | – | – | – | – | – | – | use_hindsight |
| Mixing Factor (p) | 0 | 0 | 0.4 | 0 | 0.3 | 0.8 | p |

Table 8: Hyperparameters used for training Tabular MDP on GridWorld-1D environment.

| Description | GridWorld | argument_name |
|---|---|---|
| Discount | 0.99 | discount |
| Batch size | 1 | batch_size |
| Number of epochs | 100 | n_epochs |
| Replay Buffer size | $3e^4$ | buffer_size |
| Buffer batch size | 64 | batch_size |
| Exploration factor | 0.3 | max_epsilon |
| Learning rate | 0.1 | lr |

# C  Additional Results

## C.1  Analysis of Fault Tolerance in Reinforcement Learning

Selecting an appropriate metric for comparison in RL is crucial due to the high variance in results attributed to the environment stochasticity and the stochasticity in the learning process. As such, using the average performance metric with small number of seeds can give statistically different distributions in results. This has been highlighted by the works of Henderson et al. (2018). Reliable reinforcement learning still remains an open problem. As such, using the average over all seeds might not give us a stable comparison of the algorithms considered, unless we can increase our number of seeds to large numbers such as > 100. This is computationally very expensive and such high number of trials are seldom used in practice. Instead, other approaches use the maximum of N runs to report performance. This however is not ideal since it overestimates your performance by a large amount. We can see this with the PER hyper-parameter search where we deploy the best performing hyper-parameter from the hyper-parameter grid search into an independent experiment with Top-K out of n metric. We notice that this under-performs the grid search result by a large margin. Thus, the top-1 metric is not desirable.

Instead, we propose the use of the Top-K performance metric, and the Robust Mean performance metric which we show below to be robust to noise and is closer to providing the right analysis in comparison to

average metric approach. Similar to work such as Mnih et al. (2016) which used a top-5/50 experiments, we use a top-10/25 experiments to limit the overestimation of results further.

To explain our decision choice of Top-K metric and Robust Mean metric, we consider a fault-based model for RL algorithm output and show via a simulation of this in a simple toy example why Top-K seeds or Robust Mean metric can give us a better way of inferring the best algorithm than the average, especially when the number of independent trials is low. RL algorithms usually perform well sometimes, and some other times they fail badly. This has been noted in the literature Henderson et al. (2018) and can also be seen in our experiments. Since the deviations are so large between seeds, this can be modeled as a 'fault' rather than a mean + gaussian noise. To give a better understanding, let us elaborate with a simplified scenario: Consider the following setting where we have 20 different environments. There are two algorithms: A and B. Algorithm A gives a moving average return of 0.9 50% of the time and a moving average return of 0 the remaining times (on all 20 environments). Algorithm B, on the other hand, gives a moving average return of 1 50% of the time and a moving average return of 0 the remaining times (on all 20 environments). In reality, Algorithm B performs better than Algorithm A in all 20 environments. We can conclude this with the empirical average if we extend the number of experiments to very large numbers, such as 50 seeds per environment or larger. We further extend our above analysis by adding another Algorithm C that gives a moving average return of 0.8 50% of the time and a moving average return of 0 the remaining times (on all 20 environments).

In this simplified model, we test how well the Top-K or Robust Mean metric and the average metric perform in recovering the ground truth (i.e, Algorithm B) via Monte-Carlo simulation with 500 trials. In each trial, we generate the return of each algorithm over each of the 20 environments from the model described above with 25 different random seeds. For each environment, we check the best algorithm with the average metric and the best algorithm with the Top-K metric and Robust Mean metric respectively. We compare this with the ground truth (i.e, algorithm B being the best in all 20 environments). Figure 7(a) depicts the comparison between the average metric, Top-K metric, and Robust Mean metric with respect to the ground truth. As illustrated, the Top-K metric, and Robust Mean metric is more robust to faults and is closer to the ground truth than the other metrics. We further add gaussian noise of mean 0, and standard deviation 0.2, keeping all other parameters constant. This noise is added to fault runs and standard ones. We note little difference in our results and depict the results averaged over 500 runs as depicted in Figure 7(b) Therefore, our model suggests that the Top-K metric, and Robust Mean metric is more robust to faulty runs and can help facilitate the comparison of various learning algorithms with high volatility.

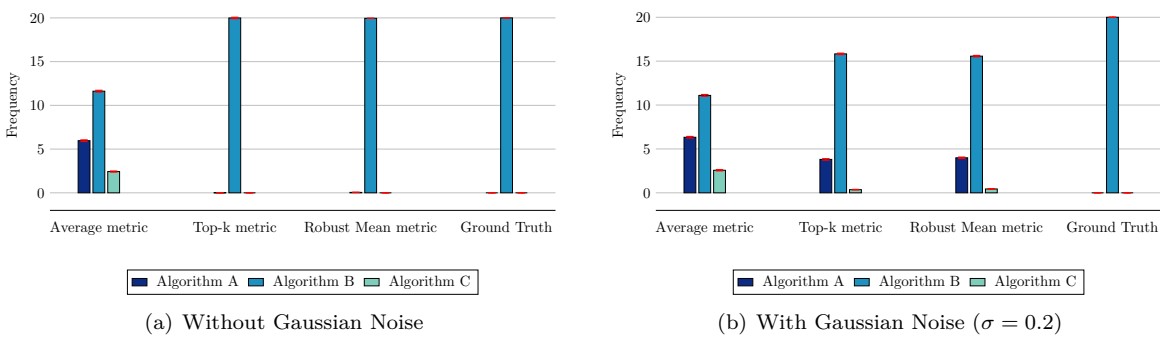

(a) Without Gaussian Noise        (b) With Gaussian Noise ($\sigma = 0.2$)

Figure 7: Analysis of Fault Tolerance Metrics.

## C.2 Grid-Search for tuning PER hyperparameters

In this section we present the results from our grid search experiment for tuning the hyperparameters for PER. We tune the bias annealing parameter ($\beta$), and the prioritization exponent ($\alpha$). We perform a robust grid search across all environments we have experimented with with the range of the priortization exponent and beta as $[0.2, 0.4, 0.6, 0.8, 1.0]$. Figure 8 illustrates the performance of PER with varying hyperparameter

We briefly summarize the findings from our grid-search experiment below:

- On environments such as CartPole , Reacher, we notice only a marginal differences between default hyper parameters ($\alpha = 0.4$, $\beta = 0.6$) and the best hyper parameters tuned for the model.

- For other environments such as HalfCheetah, Ant, FetchReach, and Walker, we notice a significant gap between the performance of PER and IER even after a thorough tuning of PER hyperparameters. For instance, IERoutperforms PER by more that 8000 in terms of average return on HalfCheetah environment. Furthemore, IER outperforms PER by almost 48 in terms of average return on FetchReach. Finally, IER outperforms PER by more than 6200 in terms of average return on Ant. On Walker, IER outperforms PER by almost 1500 in terms of average return. For these environments, we report the results with the default setting, i.e. $\alpha = 0.4$ and $\beta = 0.6$.

- Some environments such as Acrobot, Pendulum, LunarLander, Hopper, and Double-Inverted-Pendulum did show significant improvements when tuned for the prioritization exponent, and bias annealing factor. We take the best hyperparameters from the grid-search experiment, and re-run the code to compute Top-K metrics (i.e, we pick top 10 out of 25 runs). However, from our experiments, we show that even the select hyperparameter is not robust across seeds, and overall performs worse than our proposed approach of IER across all environments: Acrobot, Pendulum, LunarLander, Hopper, and Double-Inverted-Pendulum.

- For Atari environments such as Pong and Enduro, we use the default parameters recommended by Schaul et al. (2015), and do not perform a grid-search experiment.

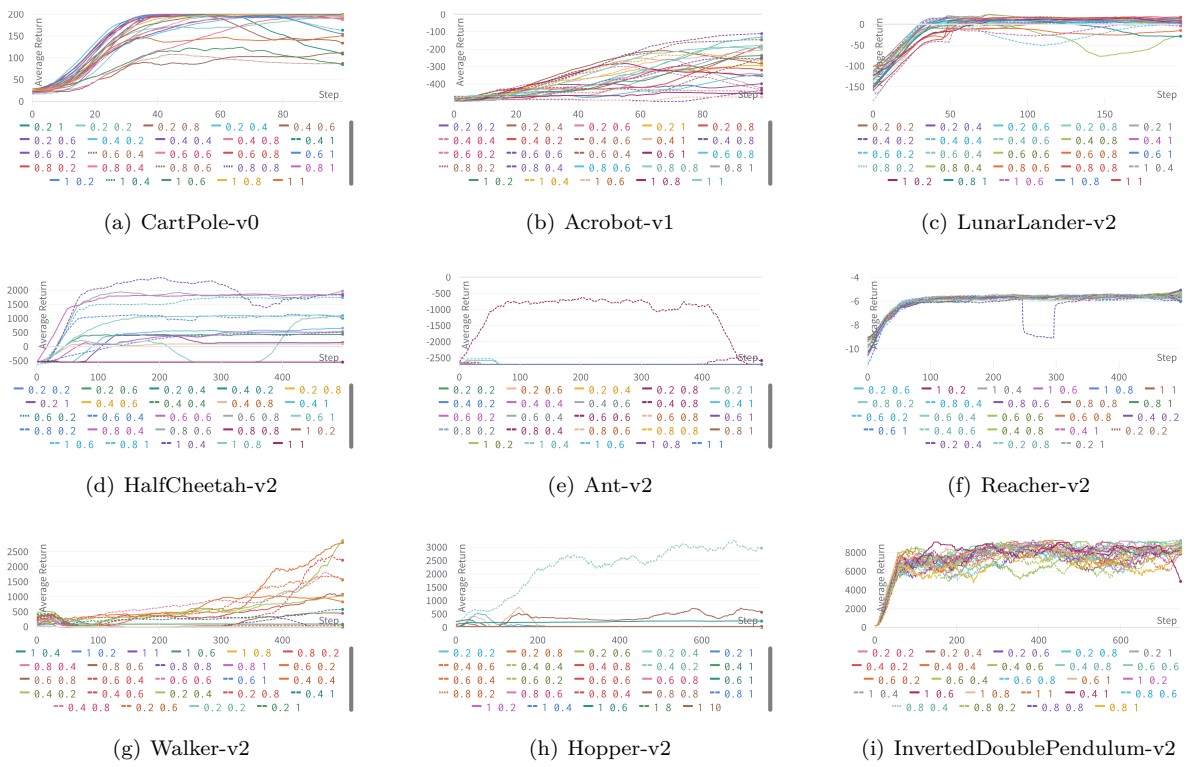

Figure 8: Grid-Search of Prioritization exponent ($\alpha$), and Bias Annealing parameter ($\beta$) respectively.

## C.3  Whole vs. Component Parts

This section briefly presents the learning curves of our models on three different sampling schemes: IER, OER and RER.

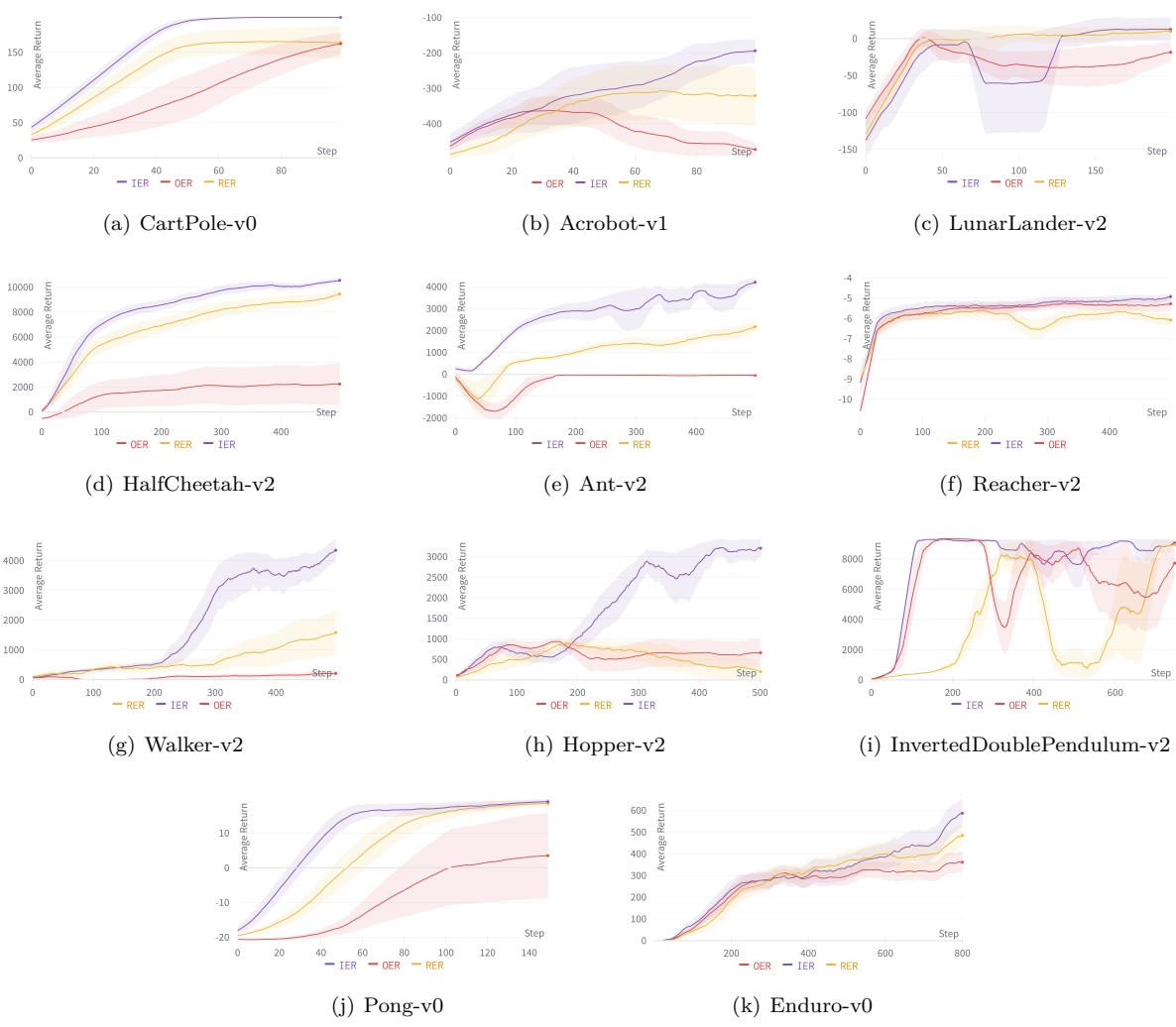

Figure 9: Ablation study of OER, RER and IER.

## C.4 Buffer Batch size sensitivity of IER

This section briefly presents the sensitivity to the buffer batch size hyperparameter for our proposed approach (IER). To analyze this, we run our experiments on the CartPole environment with varying batch size of the range 2-256. Table 9 and Figure 10 depict the buffer batch size sensitivity results from our proposed sampler.

## C.5 How important is sampling pivots?

This section briefly presents the ablation study to analyze the importance of sampling "surprising" states as pivots. As a baseline, we build a experience replay where these pivots are randomly sampled from the buffer. The "looking back" approach is used to create batches of data. For nomenclature, we refer to our proposed approach (IER) to use the "TD Metric" sampling of pivots, and the baseline that uses "Uniform" sampling of pivots. Table 10 and Figure 11 depict the buffer batch size sensitivity results from our proposed sampler.

## C.6 How important is "looking back"?

This section briefly presents the ablation study to analyze the importance of "looking back" after sampling pivots. As a baseline, we build a experience replay where we sample uniformly instead of looking back. For

Table 9: Buffer Batch size sensitivity of IER on the CartPole environment.

| Buffer Batch Size | Average Reward |
|---|---|
| 2 | 126.69 $_{\pm\ 41.42}$ |
| 4 | 192.33 $_{\pm\ 13.29}$ |
| 8 | 181.27 $_{\pm\ 32.13}$ |
| 16 | 199.24 $_{\pm\ 1.32}$ |
| 32 | **199.99** $_{\pm\ 0.001}$ |
| 64 | 199.83 $_{\pm\ 0.31}$ |
| 128 | 193.23 $_{\pm\ 10.08}$ |
| 256 | 179.95 $_{\pm\ 18.94}$ |

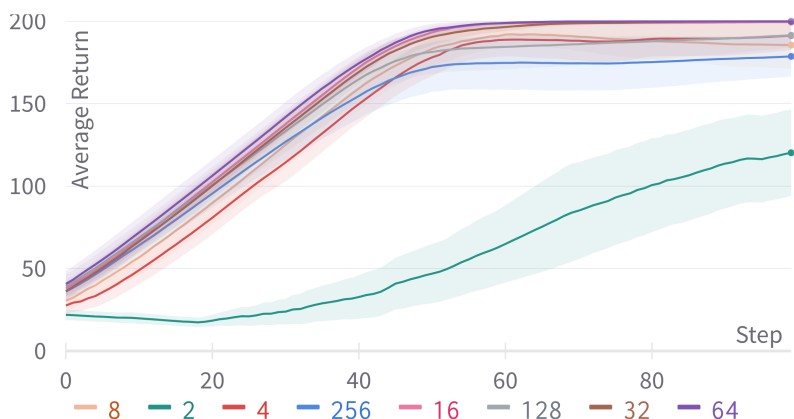

Figure 10: Buffer batch size sensitivity of IER sampler on the CartPole Environment

Table 10: Importance of sampling pivots in our proposed approach (IER) on the CartPole environment.

| Sampling Scheme | Average Reward |
|---|---|
| TD Metric (IER) | **199.83** $_{\pm\ 0.31}$ |
| Uniform (IER) | 136.71 $_{\pm\ 19.59}$ |

nomenclature, we refer to our proposed approach (IER) to use the "Looking Back" approach (similar to IER), and the baseline that uses "Uniform" approach. We refer to these two approaches as possible filling schemes, i.e. fill the buffer with states once the pivot state is sampled.

Table 11 and Figure 12 depict the buffer batch size sensitivity results from our proposed sampler.

Table 11: Importance of looking back in our proposed approach (IER) on the CartPole environment.

| Filling Scheme | Average Reward |
|---|---|
| Looking Back (IER) | **199.83** $_{\pm\ 0.31}$ |
| Uniform (IER) | 182.5 $_{\pm\ 23.49}$ |

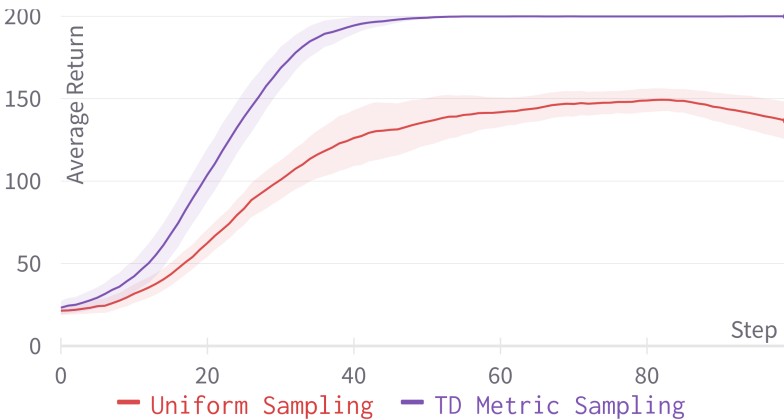

Figure 11: Ablation study of Importance Sampling of IER sampler on the CartPole Environment. Here "Uniform Sampling" denotes the uniformly random sampling of pivots, and "TD Metric Sampling" denotes our proposed approach (IER).

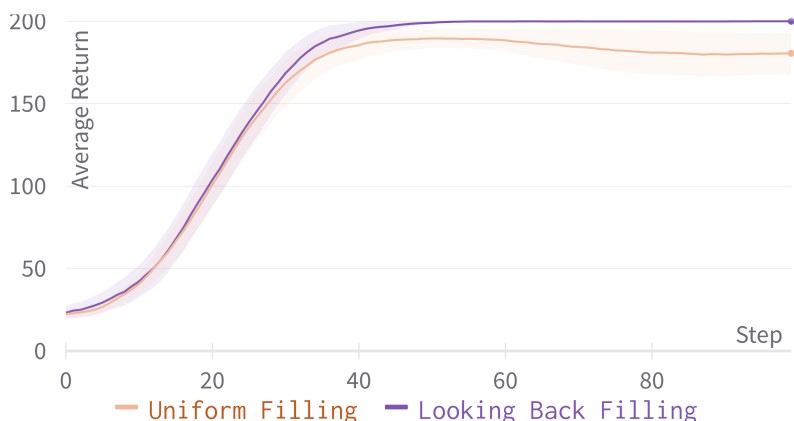

Figure 12: Ablation study of Filling Scheme of IER sampler on the CartPole Environment. Here "Uniform Filling" denotes the uniformly random sampling of states to fill after sampling the pivot state, and "Looking Back Filling" denotes our proposed approach (IER).

## D Ablation study of Temporal effects

This section studies the ablation effects of going temporally forward and backward once we choose a pivot/surprise point. Furthermore, Figure 13 depicts the learning curves of the two proposed methodologies. The forward sampling scheme is worse in most environments compared to the reverse sampling scheme.

## E Sparsity and Rewards of Surprising States

### E.1 Surprising States Have Large Rewards

In this section, we study the "learning from sparse reward" intuition provided in Section 4 – i.e., we want to check if the states corresponding to large TD error correspond to states with large (positive or negative) rewards. To test the hypothesis, we consider a sampled buffer and plot the TD error of these points in the buffer against the respective reward. Figure 3 shows the distribution of TD error against reward for the

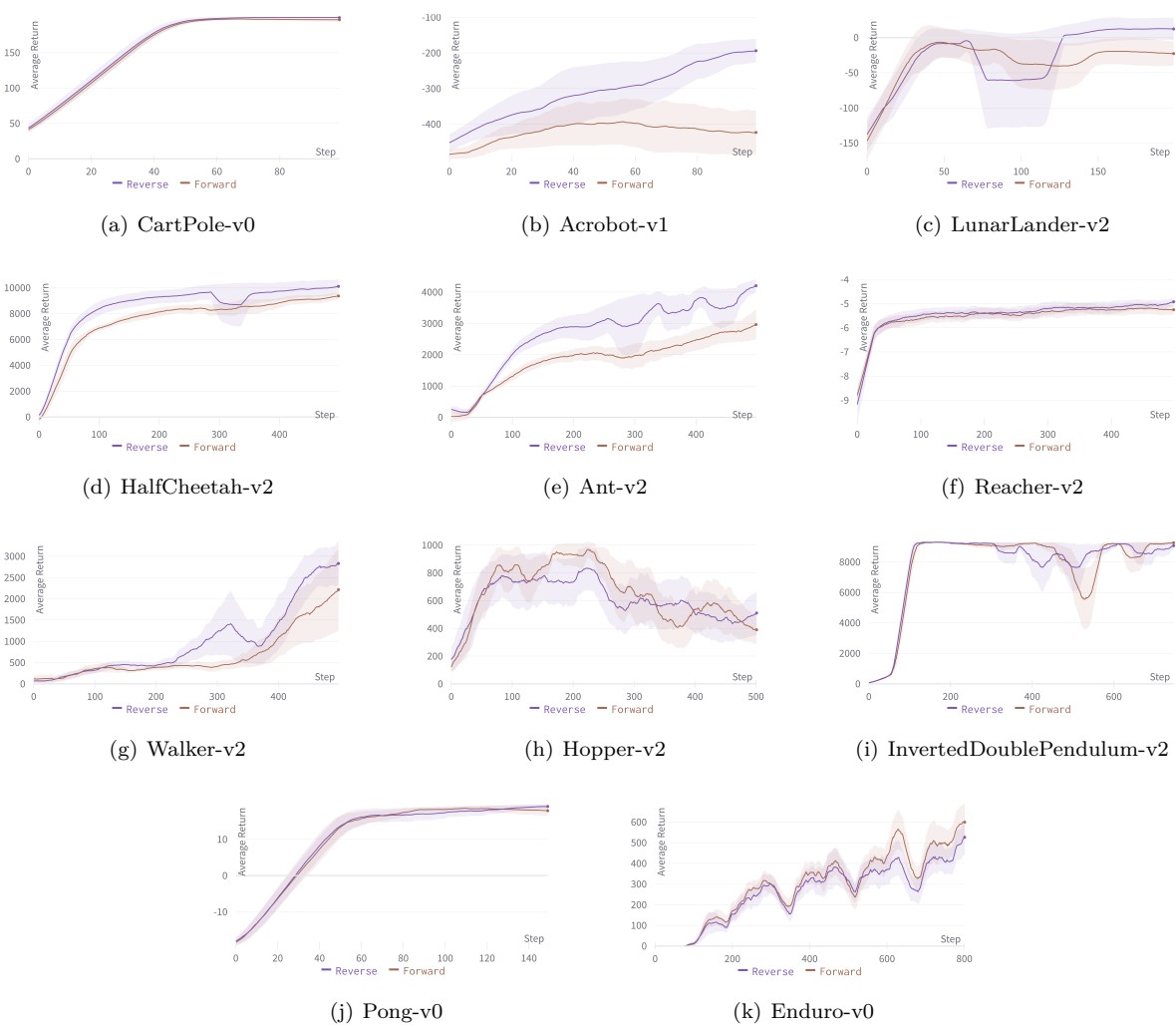

Figure 13: Ablation study of the effects of the temporal structure on the performance of the agent.

sampled buffers in the Ant environment. We see that high reward states (positive or negative) also have higher TD errors. Therefore, our algorithm picks large reward states as endpoints to learn in such environments.

### E.2 Surprising States are Sparse and Isolated

Figure 14 and Figure 15 depict the distribution of "surprise"/TD error in a sampled batch for CartPole and Ant environments respectively. These two figures help show that the states with a large "surprise" factor are few and that even though the pivot of a buffer has a large TD error, the rest of the buffer typically does not.

Figure 14(d) and Figure 15(d) show a magnified view of Figure 14(c) and Figure 15(d) where the pivot point selected is dropped. This helps with a uniform comparison with the remaining timesteps within the sampled buffer. Again, we notice little correlation between the timesteps within the sampled buffer.

## F  Reverse Experience Replay (RER)

This section discusses our implementation of Reverse Experience Replay (RER), which served as a motivation for our proposed approach. The summary of the RER approach is shown in Figure 17. Furthermore, an overview of our implemented approach to RER is described briefly in Algorithm 2.

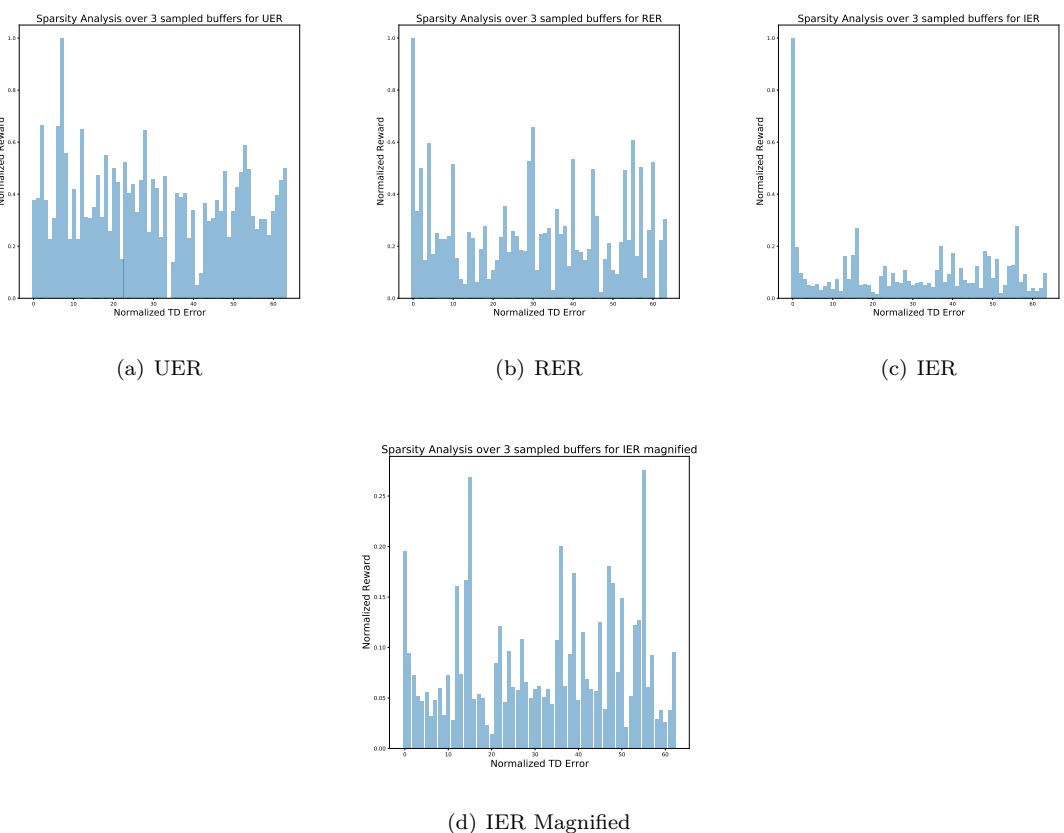

(a) UER

(b) RER

(c) IER

(d) IER Magnified

Figure 14: Normalized TD Error ("Surprise factor") of each timestep over three different sampled buffers on the CartPole environment. Best viewed when zoomed.

---

**Algorithm 2:** Reverse Experience Replay

**Input:** Data collection mechanism $\mathbb{T}$, Data buffer $\mathcal{H}$, Batch size $B$, grad steps per Epoch $G$, number of episodes $N$, learning procedure $\mathbb{A}$

$n \leftarrow N$;
$P \leftarrow \mathsf{len}(\mathcal{H})$ ;                             // Set index to last element of Buffer $\mathcal{H}$
**while** $n < N$ **do**
    $n \leftarrow n + 1$;
    $\mathcal{H} \leftarrow \mathbb{T}(\mathcal{H})$ ;                          // Add a new episode to the buffer
    $g \leftarrow 0$;
    **while** $g < G$ **do**
        **if** $P - B < 0$ **then**
            $P \leftarrow \mathsf{len}(\mathcal{H})$ ;                  // Set index to last element of Buffer $\mathcal{H}$
        **else**
            $P \leftarrow P - B$;
        **end**
        $D \leftarrow \mathcal{H}[P - B, P]$ ;                 // Load batch of previous $B$ samples from index $P$
        $g \leftarrow g + 1$;
        $\mathbb{A}(D)$;                               // Run the learning algorithm with batch data $D$
    **end**
**end**

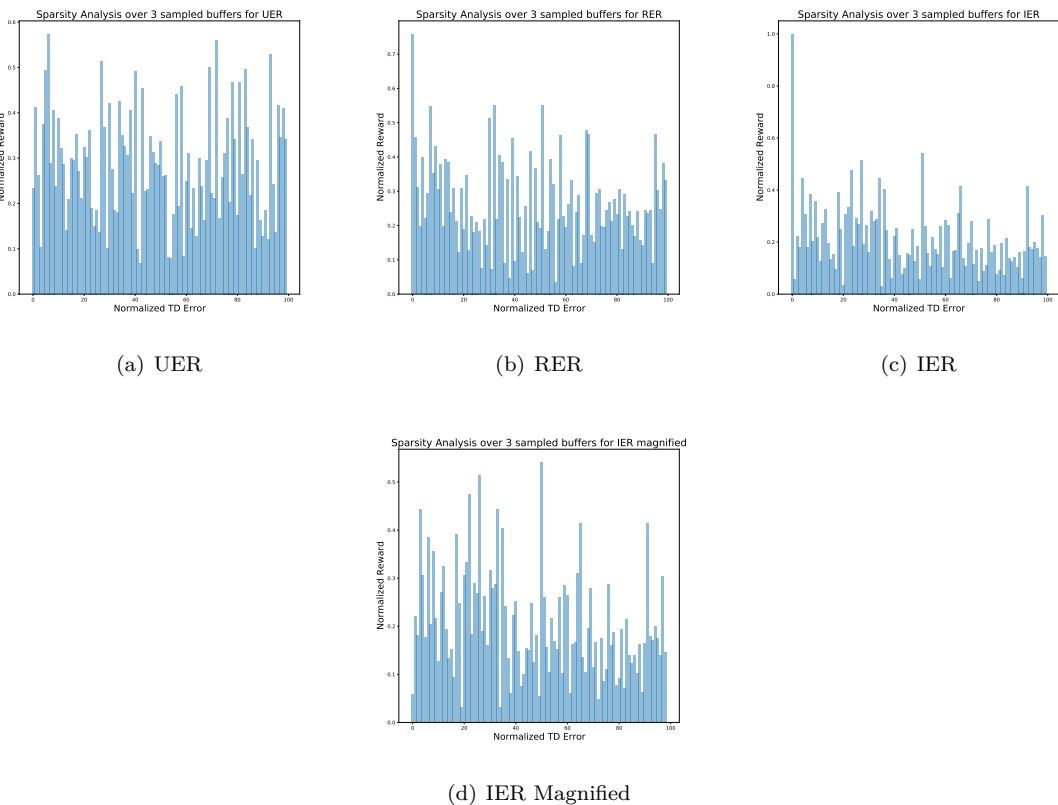

(a) UER  (b) RER  (c) IER

(d) IER Magnified

Figure 15: Normalized TD Error ("Surprise factor") of each timestep over three different sampled buffers on the Ant environment. Best viewed when zoomed.

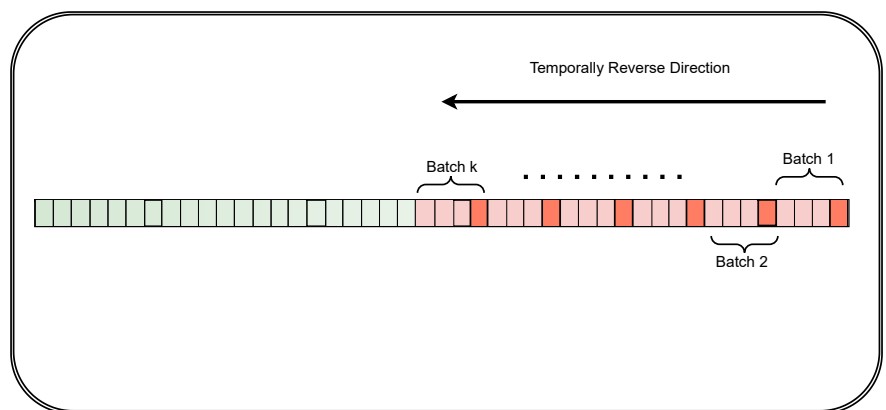

Figure 16: An illustration of Reverse Experience Replay (RER) when selecting $k$ batches from the Replay Buffer.

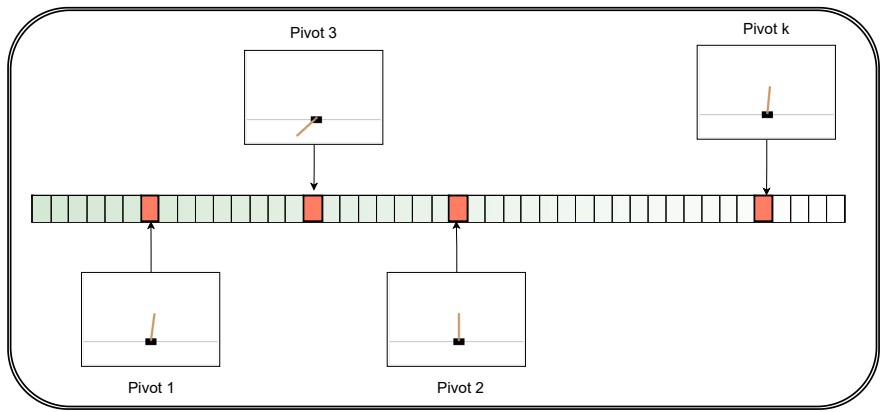

Figure 17: An illustration of Optimistic Experience Replay (RER) when selecting $k$ pivots to form a batch from the Replay Buffer.

