# OpenReview forum: "Introspective Experience Replay: Look Back When Surprised"
_TMLR — Accepted by TMLR_

### Review · Reviewer_TQ7y · 2023-10-04

**Summary Of Contributions:**

This paper presents an advanced algorithm known as the Introspective Experience Replay (IER), designed to enhance experience replay-based sampling in Reinforcement Learning (RL). The IER algorithm operates by selectively sampling batches of data points preceding unexpected events. The method identifies the top 'k' pivot points within a data buffer, ranking these points based on their Temporal Difference (TD) error. The algorithm subsequently generates data batches by selecting points that temporally precede these pivot points.

The findings of this research are primarily conveyed in a pragmatic and intuitive manner. This is achieved through substantial experimentation across various commonly studied RL environments. These encompass traditional control environments, Box-2D environments, Mujoco environments, and Atari environments.

**Audience:**

Yes

**Broader Impact Concerns:**

No significant broader impact.

**Claims And Evidence:**

Yes

**Requested Changes:**

I am recommedning fixing the above issues, including:
1) Rectify the inconsistent notation.
2) Incorporate theoretical results.
3) Address the incomplete experiments and thoroughly re-examine the outcomes to ensure accuracy and reliability.
3) Include more empirical results if possible.

**Strengths And Weaknesses:**

### Strengths
1. The proposed algorithm is easy to follow and clearly presented.
2. The comprehensive empirical results effectively highlight the resilience and versatility of the Introspective Experience Replay (IER) algorithm.
3. The paper is well-structured and written, with many details provided to aid readers in understanding the principal contributions.

### Weaknesses
1. **Technical Issues**. This paper primarily employs a episodic reinforcement learning setting that is grounded in a non-stationary Markov Decision Process (MDP). Consequently, both the policy and the value function should inherently be time-dependent. However, it appears that the algorithmic constructs and notations in Section 3.1 do not align with these stipulated settings.

2. **Lack of theoretical Insights**. The paper appears to underscore that one of the primary limitations of previous works, such as Prioritized Experience Replay (PER), stems from their limited theoretical results. Given that IER is essentially a combination of OER and RER, and both of which have been theoretically examined - it is unclear why a similar analysis for IER would present significant challenges. Considering the authors claims that "further theoretical investigation is needed" for IER, I believe this work remains incomplete without the inclusion of such theoretical results. I am assuming the authors possess a robust theoretical background in Reinforcement Learning (RL), particularly in experience replay-based sampling, and I am confident that they can accomplish it.

3. **Concerns over Emprical Findings.** Based on the key findings of this paper (as illustrated in Figure 5), it appears that the training process was not fully completed in several environments. Specifically, the training curves for (b) Acrobot-v1, (g) Walker-v2, and (k) Enduro-v0 are still on an ascending trajectory and have yet to stabilize by the end of the training period. Additionally, I am curious why the Top-k results occasionally underperform compared to the Robust Mean results. This counterintuitive outcome merits further investigation to understand the underlying causes.

4. **Novelty**. After reviewing the RER and OER algorithm, it seems the enhancements made by the proposed IER algorithm are relatively minor, particularly given the absence of theoretical foundation supporting the significance of this update, although my impression could be subjective and incomplete. I leave the decision to AC.

Minor Concerns:

Atari is a big benchmark and this paper only picks two (Pong and Enduro) out of 57 in total environments. I am wondering the reason behind this selection.

---

> ### Author Response · Authors · 2023-11-27
> **Response to Reviewer TQ7y (Part 1/2)**
>
> We thank the reviewer for the kind comments and insightful questions. We are happy to hear that the reviewer finds the overall idea well presented and clear, empirical evaluations comprehensive, and paper well written. Please refer below for the responses to specific questions.
>
> ---
>
> > Technical issues.
>
> - Thank you for pointing this out. We agree that the MDPs are not stationary (therefore, transition matrix and rewards are time dependent). We have made changes to indicate that the transitions and rewards are time dependent. However, standard algorithms implement stationary MDP algorithms to solve these problems (see links to the used code bases below). In order to be consistent with the implemented algorithms we continue to use time invariant notations for policy and value functions.
> [DQN](https://github.com/rlworkgroup/garage/blob/master/src/garage/torch/algos/dqn.py#L279), [TD3](https://github.com/rlworkgroup/garage/blob/master/src/garage/torch/algos/td3.py#L325).
>
> ---
>
> > Lack of theoretical Insights.
>
> - Below we point out the precise reasons why the current scenario requires fundamentally new techniques compared to prior works on experience replay. We also explain how this is beyond the scope of the current work and makes for a separate research project in itself.
>
>   RL algorithms can be on-policy control algorithms or off-policy algorithms. In off policy algorithms, there is a fixed behavior policy mapping the state space to a random action with good coverage over the state space. This policy is fixed and cannot be changed by the algorithm.  The task here is to learn a near optimal policy. The main challenge in this setting  is to overcome the mixing time dependence and divergence of the Q learning algorithm due to “the deadly triad” of bootstrap, function approximation and off policy learning (see “Reinforcement Learning, An Introduction” by Barto and Sutton). The prior works on RER consider this scenario. In fact, it is easy to see that with tabular RL, deterministic MDP and a large buffer, OER reduces the TD error in the L infinity norm the most in every iteration. RER, as pointed out by prior works, removes the mixing time dependence.
>
>   However, the scenario considered in our work  is that of on-policy control where the main problem one encounters is exploration i.e, we deploy new policies where the goal is to explore new states while also learning the optimal policy. In contrast with off policy RL, OER can fail badly as demonstrated by our experiments. RER also does not perform very well in comparison to other algorithms even in simple settings.
>
>   While we can easily write down theorems for IER in the off-policy RL setting based on prior works, we believe that this does not serve any scientific purpose because it does not explain the behavior of any of these experience replay algorithms in the on-policy control setting. A theoretical treatment of this algorithm is something we are exploring currently. For the reasons explained above we believe it involves multiple novel technical ideas - such as how IER learns good trajectories to high reward states - which are best explored in a separate manuscript.
>
> ---
>
> > Concerns over Emprical Findings.
>
> - Thanks for pointing this out. We have re-run these baselines for a longer number of steps, and illustrate a similar trend and updated the numbers in the draft.
>
> ---
>
> > Additionally, I am curious why the Top-k results occasionally underperform compared to the Robust Mean results. This counterintuitive outcome merits further investigation to understand the underlying causes.
>
> - After a detailed check, we could not find any points in the graph where top-k underperforms the robust mean. We are happy to consider any specific experiments the reviewer wants to point out. (Note: we did not have a table for top-k in our manuscript)
>
> ---
>
> > Novelty
>
> - We would like to respectfully disagree with this statement. We want to point out that IER vastly outperforms both OER and RER (see Figure 9), while also achieving better performance compared to UER and PER over a wide range of environments. Thus, we believe that our method is of interest to the RL community.
>
> ---

---

> > ### Author Response · Authors · 2023-11-27
> > **Response to Reviewer TQ7y (Part 2/2)**
> >
> > > Atari is a big benchmark and this paper only picks two (Pong and Enduro) out of 57 in total environments. I am wondering the reason behind this selection.
> >
> > - Our intention was to demonstrate the wide applicability of our method beyond Atari games by experimenting on a diverse set of environments. We chose 11 environments, including a few Atari games that have been previously used to support the efficacy of other reinforcement learning algorithms such as UER and PER, as well as other classes of environments, such as Classic Control, Box 2D, and Mujoco.
> >
> >   We have attempted to reproduce various atari environments through multiple repositories including Stable Baselines and Garage. We included the two (Pong and Enduro) we were able to reproduce using standard code bases used by RL researchers.
> >
> >   Stable Baselines baseline runs of DQN on Q*Bert, Breakout, Seaquest, BeamRider and SpaceInvaders (https://github.com/DLR-RM/stable-baselines3) averaged over 3 seeds are reported below.
> >
> > | Environment   | Average Reward (ours)  | Average Reward (DQN paper) |
> > |---------------|------------------|------------------------|
> > | Q*bert        | 690 ± 33.18      | 1952                   |
> > | Breakout      | 25.07 ±  4.05    | 168                    |
> > | Seaquest      | 262 ±  27.87     | 1705                   |
> > | Beamrider     | 541.33 ±  154.76 | 4092                   |
> > | SpaceInvaders | 256.67 ±  10.07  | 581                    |
> >
> > These are still significantly different from the numbers reported in the original DQN paper, and we are still working on reproducing DQN on other Atari benchmarks.
> >
> >
> >
> > ---
> >
> > We hope that the rebuttal clarifies the questions raised by the reviewer. We would be very happy to discuss any further questions about the work, and help facilitate the acceptance of the paper if the reviewer’s concerns, and requested changes are adequately addressed.

---

> ### Comment · Action_Editors · 2023-12-10
> **Discussion**
>
> Dear reviewer,
>
> Have the authors properly addressed your concerns?
>
> Thank you!

---

> > ### Comment · Reviewer_TQ7y · 2023-12-17
> > **Thank you for the rebuttal**
> >
> > Thanks for your response. I think that the rebuttal has addressed the majority of the concerns raised. However, it should be noted that there is a lack of novelty and theoretical depth in the work. Apart from that, I am fine with the current draft, and I have no additional questions at this point.

---

> > > ### Author Response · Authors · 2023-12-18
> > >
> > > We are glad the reviewer found the majority of the concerns addressed.
> > > We want to stress that the novelty aspect of this work comes from the amalgamation of RER and OER, which was not proposed earlier. We also show that IER outperforms either RER or OER significantly, as stressed earlier.
> > > Furthermore, due to the above-mentioned open-ended questions and research problems in studying the algorithm theoretically, we leave this as a future work to be best explored in a separate manuscript.
> > >
> > > Again, thank you for your time and effort in reviewing our work.

---

### Review · Reviewer_iPCH · 2023-10-12

**Summary Of Contributions:**

This paper presents an innovative approach called Introspective Experience Replay (IER), which introduces a strategy for experience replay in machine learning. The IER method first selects a set of pivotal samples and then retrieves samples that precede these pivot points to construct a batch of data. Extensive experiments across diverse domains underscore not only the overall effectiveness of this method but also the contributions of its individual components.

**Audience:**

Yes

**Claims And Evidence:**

No

**Requested Changes:**

* As IER is a combination of OER and RER, it would be better if the authors could include more detailed introductions about those two methods in the main text. For example, it might be a good idea to add figures like Fig.1 for OER and RER.
* Section 4.1 appears somewhat perplexing. It would be beneficial if the authors could provide further elaboration on the specific intent and significance of this subsection.
* On page 5, the authors mention that IER(F) might choose samples from the next episode. However, I think it is more of an implementation bug. We can easily restrict it within the same episode. Similarly, IER might also select samples from the previous episode. Please consider fixing it in the ablation experiments.
* Page 12: how is the variant where the points are randomly sampled implemented? Are the points randomly sampled around pivots or over the entire buffer?
* Table 4: could the authors give some comments on the reduced time efficacy in MuJoCo?
* See Weaknesses for other changes.

Minor changes:
* There is no verb in the first sentence under Section 4.
* Page 9: "the task to infer" -> "the task is to infer"

**Strengths And Weaknesses:**

Strength:
* The writing is in general clear.
* The proposed IER method is simple and easy to understand and implement.

Weaknesses:
* While IER builds upon the foundations of RER, it remains somewhat unclear how IER benefits from a comparable level of theoretical underpinning.
* In the abstract, the authors state that IER is introduced to tackle issues related to "sub-optimal convergence" and "high sensitivity to initial conditions." Nevertheless, there remains some ambiguity in how IER has specifically and effectively mitigated these two challenges. Furthermore, it's worth noting that the term "convergence" is typically employed within a theoretical context, yet the authors seem to apply it to describe empirical performance in this case.
* The connection between high TD errors and large reward states is not surprising. I didn't quite get the point of highlighting it. Besides, the authors state that "IER picks pivots which are the states with large (positive or negative) rewards, enabling effective learning.". Nonetheless, it appears that this outcome isn't a distinctive characteristic of IER but rather an effect of the sampling strategy (focusing on states with high TD errors). It's worth noting that OER also employs a similar strategy.

---

> ### Author Response · Authors · 2023-11-27
> **Response to Reviewer iPCH (Part 1/2)**
>
> We thank the reviewer for the kind comments and insightful questions. We are happy to hear that the reviewer finds the overall writing clear, and proposed idea simple and easy to implement. Please refer below for the responses to specific questions.
>
> ---
>
> > While IER builds upon the foundations of RER, it remains somewhat unclear how IER benefits from a comparable level of theoretical underpinning.
>
> - We partially agree with the reviewer that IER does not enjoy a similar level of theoretical underpinning as RER. We acknowledge this in the manuscript and state that a sharp theoretical analysis is necessary towards this end. In this work we use RER as a guiding principle. Through empirical evaluations, we show that combining this with OER gives an algorithm which vastly outperforms either of them and achieves SOTA performance. We are currently working on a separate project trying to theoretically analyze IER. We refer to the [response to Reviewer TQ7y](https://openreview.net/forum?id=vWTZO1RXZR&noteId=StsLj7h2Fv) to explain the technical challenges which we encounter in this process.
>
> ---
>
> > In the abstract, the authors state that IER is introduced to tackle issues related to "sub-optimal convergence" and "high sensitivity to initial conditions." Nevertheless, there remains some ambiguity in how IER has specifically and effectively mitigated these two challenges....
>
> - We agree that sub-optimal convergence can be confusing when seen in a theoretical context. We wanted to refer to the “the asymptotic performance/ accuracy of typically deployed ER algorithms is not the best possible with the given architecture”. Our algorithm achieves better performance than these methods, thus improving the “convergence”. We want to note that convergence is also used in ML practice in situations where learning curves do not improve with further training, and the model is said to have converged. We have clarified the term in the Introduction as suggested . “High seed sensitivity” refers to PER  which we empirically show in the paper: We pick hyperparameters for PER based on grid search but the best values obtained from grid search do not give us the similar values when these hyperparameters are chosen for multiple runs. (see Appendix C.2)
>
> ---
>
> > The connection between high TD errors and large reward states is not surprising....
>
> - The connection between high TD errors and large rewards is indeed known, and this intuition is also applied in replay buffers such as OER, PER as noted by the reviewer. What is indeed a distinctive characteristic of IER is to focus on those transitions right before a surprising point. We highlighted this fact as a first step of the justification for looking back from the high TD error state. The second part of the justification is that (Figure 3, Appendix E), the states temporally preceding the high reward states are not sampled often when the sampling strategy focuses on TD error alone since they tend to have low TD error.
>
>    Expanding more on these issues:
>
>   - We first note that OER picks exclusively such high TD error points, which can lead to poor performance (as noted in the PER paper [1], where this is referred to as the `bias’ error). The solution proposed in the PER paper is a sophisticated sampling strategy, which can be highly seed sensitive as noted above in our response.
>
>   - Our work obtains improved performance by additionally considering the temporal dynamics inherent to RL environments: That is, the states which occurred before the high reward goal states hold the key to reaching the goal states. Thus we ensure that such states are fed into the learning algorithm through the replay buffer. (add section reference here) We also show that such states are not sampled very frequently if only the TD error criterion is used since these prior states have low TD error.
>
> [1] Tom Schaul, John Quan, Ioannis Antonoglou, and David Silver. Prioritized experience replay. arXiv preprint arXiv:1511.05952, 2015.
>
> ---
>
> > As IER is a combination of OER and RER, it would be better if the authors could include more detailed introductions about those two methods in the main text. For example, it might be a good idea to add figures like Fig.1 for OER and RER.
>
> - We have detailed how RER and OER work in Section 2. Furthermore, we also have a figure of RER in the Appendix (see Figure 16). We have also added a figure illustrating OER in the Appendix now (see Figure 17). We have chosen to add these in the appendix due to space limitations. However we can try to move it to the main paper if the reviewer believes this adds value to the paper.
>
> ---

---

> > ### Author Response · Authors · 2023-11-27
> > **Response to Reviewer iPCH (Part 2/2)**
> >
> > > Section 4.1 appears somewhat perplexing. It would be beneficial if the authors could provide further elaboration on the specific intent and significance of this subsection.
> >
> > - The didactic toy example presented in Section 3.2 is a simpler problem by design. Here, the trap state in the Gridworld-1D: it is not really a "trap" state, and just a negative reward. Similar chain problems (with increased hardness of reaching the goal state) were used in the Quota paper [1], which studies explorations and used previously to compare other replay buffer algorithms in literature. The intent of these experiments is to intuitively understand and visualize how each of these algorithms work in a simple environment. This also allows us to gain a mental picture of the kind of data points chosen by each of these ER algorithms.
> >
> >
> > [1] Shangtong Zhang and Hengshuai Yao. Quota: The quantile option architecture for reinforcement learning. In Proceedings of the AAAI Conference on Artificial Intelligence, volume 33, pp. 5797–5804, 2019.
> >
> > ---
> >
> > > On page 5, the authors mention that IER(F) might choose samples from the next episode. However, I think it is more of an implementation bug. We can easily restrict it within the same episode. Similarly, IER might also select samples from the previous episode. Please consider fixing it in the ablation experiments.
> >
> > - We agree with the reviewer regarding this point. This point was meant to highlight the following fundamental conceptual difference between  IER and IER (F):
> > IER finds high reward states and picks states prior to that which contain information about how to reach the high reward state (see Section 4 (“Propogation of Sparse Rewards”), Figure 3, and Appendix E). IER (F) picks states in the future, which might not contain information about reaching such high reward states.
> >
> >   We wanted the sentence in question to illustrate the markov property: that the future states could be in another episode, which do not contain any information about how to reach the prior high reward states. We have also modified Section 3 (page 5) to alleviate any further confusion surrounding the topic.
> >
> > ---
> >
> > > Page 12: how is the variant where the points are randomly sampled implemented? Are the points randomly sampled around pivots or over the entire buffer?
> >
> > - The points are randomly sampled from the entire buffer in this case. We have also added this clearly in Page 12 to avoid any future confusion.
> >
> > ---
> >
> > > Table 4: could the authors give some comments on the reduced time efficacy in MuJoCo?
> >
> > - We briefly touch upon this point in the Speedup paragraph in Section 6 (modulo mentioning MuJoCo). We believe that as the network becomes wider and deeper, our approach does have a higher overhead (especially computing TD error which we use in the IER formulation). Since Mujoco environments make use of a much larger network, we expect the computation time of the surprise factor for each state in the buffer to be much longer. One way to avoid such an overhead is to only compute the surprise factor once in `k` number of steps. We will further expand on this in the final version as this requires some re-writing due to page constraints.
> >
> > ---
> >
> > We hope that the rebuttal clarifies the questions raised by the reviewer. We would be very happy to discuss any further questions about the work, and help facilitate the acceptance of the paper if the reviewer’s concerns, and requested changes are adequately addressed.

---

> ### Comment · Action_Editors · 2023-12-10
> **Discussion**
>
> Dear reviewer,
>
> Have the authors properly addressed your concerns?
>
> Thank you!

---

> ### Comment · Reviewer_iPCH · 2023-12-16
> **Thank you for the rebuttal**
>
> I appreciate the authors's efforts in providing a detailed response to my comments. While most of my concerns have been addressed, I think the authors should be very cautious when describing IER and avoid statements like "IER is built upon the theoretically sound RER". This is misleading and seems to suggest that "IER enjoys similar theoretical guarantees", which is however not true.

---

> > ### Author Response · Authors · 2023-12-17
> > **Thank you**
> >
> > We are glad we have cleared most of your prior concerns. As suggested by the reviewer, we have also addressed the misleading line to avoid any confusion with the readers. We hope the updated draft reflects these changes.

---

> > > ### Comment · Reviewer_iPCH · 2023-12-17
> > > **Please update abstract**
> > >
> > > Can you also update the abstract to reflect this change?

---

> > > > ### Author Response · Authors · 2023-12-18
> > > >
> > > > The abstract has also been updated to reflect this change. Thanks!

---

### Review · Reviewer_HEA6 · 2023-11-17

**Summary Of Contributions:**

This paper presents a novel method for sampling data in the replay buffer for faster convergence of reinforcement learning algorithms. This paper provide detailed survey of existing methods and leverage simple examples to illustrate the benefit of the proposed method. Experiments in simulated domains also demonstrate the effectiveness of the proposed method and how it compares with baselines.

**Audience:**

Yes

**Broader Impact Concerns:**

This work presents a technique for improving RL and does not raise any immediate ethical concerns.

**Claims And Evidence:**

Yes

**Requested Changes:**

The authors should take a closer look and proof read the paper. E.g. section 4 first sentence is missing a verb.

It would be nice for the authors to group task by properties and discuss what kind of tasks would benefit the most from IER.
More extensive empirical evidence would be useful for evaluating the real impact of IER, since there isn't any theoretical analysis.

**Strengths And Weaknesses:**

Strengths:
- This paper is well written with clarity and spent a good effort explaining and comparing with baseline methods in the literature. I find it very easy to follow despite that I am not familiar with the research in this field.
- The proposed method is simple (in a good way), building upon reverse experience replay, and intuitive. The authors did a good job providing the intuition and investigating the benefit of the proposed method by case study in toy domains.

Weaknesses:
- The proposed method does not seem to outperform baselines by much. It might be desirable for the authors to identify properties of task scenarios that IER is best, while pointing out where IER does not affect the performance by much. e.g. does IER work better for tasks that have bottleneck states?
- It seems HER was not used in the illustrations for understanding why it is not as good as IER. While HER is not a sampling method, it would still be useful to dig into why it would not perform as well as IER on the illustrative toy task.

---

> ### Author Response · Authors · 2023-11-27
> **Response to Reviewer HEA6**
>
> We thank the reviewer for the kind comments and insightful questions. We are happy to hear that the reviewer finds the overall idea interesting, and the paper well written. Please refer below for the responses to specific questions.
>
> ---
>
> >  Addition of HER to the illustrative toy task and comparisons against IER.
>
> - Thank you for highlighting this. We have added additional experiments comparing HER in the illustrative toy task. Our findings show that IER significantly outperforms HER on the two chain environments used in Section 4.1. We note that HER is not optimal for tabular representation learning, and would work better in the cases of functional approximation algorithms (see explanation below). Thus, the performance of IER is indeed significantly better than HER.
>   HER uses goal conditioned RL, where the agent tries to learn how to reach different, easier goals as a step to learn how to reach the true goal. This mitigates the issues that arise due to sparse rewards in such problems. This requires us to have a metric structure on the goal space/state space which allows the agent to learn how to reach the true goal by a) learning how to reach states close to the true goal (via HER)  and then b) learn how to go from these close states to the true goal. Such closeness relationships are expressed via function approximation. However, in the simple illustrations, we have used tabular RL, where such relationships such as “state 1 is close to state 2 but far from state 3” cannot be expressed. Thus we did not expect the goal conditioning provided by HER to help learning.
>
> ---
>
> > It would be nice for the authors to group task by properties and discuss what kind of tasks would benefit the most from IER. More extensive empirical evidence would be useful for evaluating the real impact of IER, since there isn't any theoretical analysis.
>
> - Based on our discussion of toy examples and motivations for IER, environments where the objective is to reach high reward goal.  An important feature which helps IER is when the states temporally preceding the high reward goal state have low reward. In this scenario, IER decides that the states immediately preceding the high reward state are important to learn and picks them into the batch. We have discussed this in detail for environments such as ant and cartpole in section 4. Along with the toy examples,  other environments which follow a similar design are best suitable for IER.
>   For instance, on Cartpole, we note a +1 reward if the pole is upright, and an immediate end in the episode if the angle if (a) the pole is more than 15 degrees from vertical; or 2) the cart moves more than 2.4 units from the center. Since there is a sudden change in the rewards, we believe that algorithms such as IER can capture the actions just preceding this reward and learn effectively. Similarly, for Acrobot, the goal is to have the free end reach a designated target height in as few steps as possible, and as such all steps that do not reach the goal incur a reward of -1. Achieving the target height results in termination with a reward of 0.
>
> ---
>
> > The authors should take a closer look and proof read the paper. E.g. section 4 first sentence is missing a verb.
>
> - Thank you for pointing this out. We have corrected it, and made multiple passes of the paper to fix any other discrepancies as well.
>
> ---
>
> We hope that the rebuttal clarifies the questions raised by the reviewer. We would be very happy to discuss any further questions about the work, and help facilitate the acceptance of the paper if the reviewer’s concerns, and requested changes are adequately addressed.

---

> ### Comment · Action_Editors · 2023-12-10
> **Discussion**
>
> Dear reviewer,
>
> Have the authors properly addressed your concerns?
>
> Thank you!

---

### Decision · Action_Editor_pUh9 · 2024-01-12

**Recommendation:** Accept with minor revision

**Comment:**

All three reviewers' recommendation are "Leaning Accept". They had concerns about the novelty, the lack of theoretical guarantees, and whether the method has significantly outperformed the alternatives. These are not serious concerns in my opinion. For example, the paper's novelty is combining two other experience replay methods (Reverse Experience Replay and Optimistic Experience Replay). How novel this combination is can only be subjectively evaluated. Since novelty is not the main criteria for TMLR, I do not put much weight on it.


The authors have improved the paper during the discussion phase. In my own reading of the paper, I found several typos and other minor issues, which I enlist:

- P1: Extra space between footnote superscript 1 and the previous word.
- P1: What is the citation for OER?
- P3: The sentence "Since RER forms mini-batches ... can be unreliable Neural approximation" is not grammatical. Also please explain what is special about NN that prevents RER to be reliable?
- P4, Algorithm 1: What's the purpose of ';' in Algorithm 1? Also they are not adjusted.
- P4, Algorithm 1: "Randomly chose" --> "Randomly choose".
- P5, 1st line: What is index `i' in P[I]?
- P9, last line: The comment that IER (F) might select batches overflowing into the next episode is confusing. This can be avoided by proper data keeping. This is mentioned by one of the reviewers too.
- P10, Figure 5: Are these results for DQN or TD3, or a combination of both?
- P10, Figure 5: There are occasionally big drops in the performance of IER, for example, in (c). Any explanation for that?
- P10-11, Figures 5 and 6: What are the shaded area? Standard error?
- P12: The first paragraph has several typos: "OER not RER", "it's variant", no closing parenthesis.
- P12: No space in "classes.As the"
- P12: The explanation why RER is not working well for neural approximation is that the loss function is highly non-convex and this hinders the working on RER. Why is that? What specific aspect of non-convexity hinders RER? This level of explanation may not be very insightful.


I believe the paper does not need another round of reviewing, but it definitely requires another careful proofreading and some minor revisions. That is why I recommend acceptance with minor revision. I am looking forward to receive the authors' revised paper soon.

**Audience:**

Yes. The suggested method (Introspective Experience Replay) is relevant to the Reinforcement Learning community.

**Claims And Evidence:**

Yes. The paper provides empirical evidence showing how the method performs and that it is often a reasonably good method. The paper does not provide theoretical guarantees, but that is acceptable.

---

> ### Author Response · Authors · 2024-01-26
> **Thank you**
>
> We thank the reviewers and AE for the kind words and comments. We have updated the draft to de-anonymize and resolve the comments left by the AE.
>
> For the comment:
>
> > P10, Figure 5: There are occasionally big drops in the performance of IER, for example, in (c). Any explanation for that?
>
> - We do not understand the origin of this phenomenon and we are investigating this from an angle of [catastrophic forgetting](https://www.tandfonline.com/doi/abs/10.1080/095400900116177).
>
> We hope that the updated draft fixes the comments raised by the AE and clarifies any questions raised by the AE.